# A FAST AND ACCURATE SPLITTING METHOD FOR OPTIMAL TRANSPORT: ANALYSIS AND IMPLEMENTATION

**Vien V. Mai** [*]        **Jacob Lindbäck** [†]        **Mikael Johansson** [†]

## ABSTRACT

We develop a fast and reliable method for solving large-scale optimal transport (OT) problems at an unprecedented combination of speed and accuracy. Built on the celebrated Douglas-Rachford splitting technique, our method tackles the original OT problem directly instead of solving an approximate regularized problem, as many state-of-the-art techniques do. This allows us to provide sparse transport plans and avoid numerical issues of methods that use entropic regularization. The algorithm has the same cost per iteration as the popular Sinkhorn method, and each iteration can be executed efficiently, in parallel. The proposed method enjoys an iteration complexity $O(1/\epsilon)$ compared to the best-known $O(1/\epsilon^2)$ of the Sinkhorn method. In addition, we establish a linear convergence rate for our formulation of the OT problem. We detail an efficient GPU implementation of the proposed method that maintains a primal-dual stopping criterion at no extra cost. Substantial experiments demonstrate the effectiveness of our method, both in terms of computation times and robustness.

## 1 INTRODUCTION

We study the discrete optimal transport (OT) problem:

$$\begin{array}{ll} \underset{X \geq 0}{\text{minimize}} & \langle C, X \rangle \\ \text{subject to} & X \mathbf{1}_n = p \quad \text{and} \quad X^\top \mathbf{1}_m = q, \end{array} \tag{1}$$

where $X \in \mathbb{R}_+^{m \times n}$ is the transport plan, $C \in \mathbb{R}_+^{m \times n}$ is the cost matrix, and $p \in \mathbb{R}_+^m$ and $q \in \mathbb{R}_+^n$ are two discrete probability measures. OT has a rich history in mathematics and operations research dating back to at least the 18th century. By exploiting geometric properties of the underlying ground space, OT provides a powerful and flexible way to compare probability measures. It has quickly become a central topic in machine learning and has found applications in countless learning tasks, including deep generative models (Arjovsky et al., 2017), domain adaptation (Courty et al., 2016), and inference of high-dimensional cell trajectories in genomics (Schiebinger et al., 2019); we refer to Peyré & Cuturi (2019) for a more comprehensive survey of OT theory and applications. However, the power of OT comes at the price of an enormous computational cost for determining the optimal transport plan. Standard methods for solving linear programs (LPs) suffer from a super-linear time complexity in term of the problem size. Such methods are also very challenging to parallelize on modern processing hardware. Therefore, there has been substantial research in developing new efficient methods for OT. This paper advances the state of the art in this direction.

### 1.1 RELATED WORK

Below we review some of the topics most closely related to our work.

**Sinkhorn method** The Sinkhorn (SK) method (Cuturi, 2013) aims to solve an approximation of (1) in which the objective is replaced by a regularized version of the form $\langle C, X \rangle - \eta H(X)$. Here, $H(X) = -\sum_{ij} X_{ij} \log(X_{ij})$ is an entropy function and $\eta > 0$ is a regularization parameter. The Sinkhorn method defines the quantity $K = \exp(-C/\eta)$ and repeats the following steps

$$u_k = p/(K v_{k-1}) \quad \text{and} \quad v_k = q/(K^\top u_k),$$

---

[*]Ericsson AB, `vien.mai@ericsson.com`
[†]KTH Royal Institute of Technology, {`jlindbac,mikaelj`}`@kth.se`

until $\|u_k \odot (Kv_k) - p\| + \|v_{k-1} \odot (K^\top u_k) - q\|$ becomes small, then returns $\operatorname{diag}(u_k) K \operatorname{diag}(v_k)$. The division $(/)$ and multiplication $(\odot)$ operators between two vectors are to be understood entry-wise. Each SK iteration is built from matrix-vector multiplies and element-wise arithmetic operations, and is hence readily parallelized on multi-core CPUs and GPUs. However, due to the entropic regularization, SK suffers from numerical issues and can struggle to find even moderately accurate solutions. This problem is even more prevalent in GPU implementations as most modern GPUs are built and optimized for single-precision arithmetic (Kirk & Wen-Mei, 2016; Cheng et al., 2014). Substantial care is therefore needed to select an appropriate $\eta$ that is small enough to provide a meaningful approximation, and large enough to avoid numerical issues. In addition, the entropy term enforces a dense solution, which can be undesirable when the optimal transportation plan itself is of interest (Blondel et al., 2018). We mention that there has been substantial research in improving the performance of SK (Alaya et al., 2019; Altschuler et al., 2017; Blondel et al., 2018; Lin et al., 2019). Most of these contributions improve certain aspects of SK: some result in more stable but much slower methods, while others allow to produce sparse solutions but at a much higher cost per iteration. Some of these sophisticated changes make parallelization challenging due to the many branching conditions that they introduce.

**Operator splitting solvers for general LP** With a relatively low per-iteration cost and the ability to exploit sparsity in the problem data, operator splitting methods such as Douglas-Rachford splitting (Douglas & Rachford, 1956) and ADMM (Gabay & Mercier, 1976) have gained widespread popularity in large-scale optimization. Such algorithms can quickly produce solutions of moderate accuracy and are the engine of several successful first-order solvers (O'donoghue et al., 2016; Stellato et al., 2020; Garstka et al., 2021). As OT can be cast as an LP, it can, in principle, be solved by these splitting-based solvers. However, there has not been much success reported in this context, probably due to the *memory-bound* nature of large-scale OTs. For OTs, the main bottleneck is not floating-point computations, but rather time-consuming memory operations on large two-dimensional arrays. Even an innocent update like $X \leftarrow X - C$ is more expensive than the two matrix-vector multiplies in SK. To design a high-performance splitting method, it is thus crucial to minimize the memory operations associated with large arrays. In addition, since most existing splitting solvers target general LPs, they often solve a linear system at each iteration, which is prohibitively expensive for many OT applications.

**Convergence rates of DR for LP** Many splitting methods, including Douglas-Rachford, are known to converge linearly under strong convexity (see e.g. Giselsson & Boyd (2016)). Recently, it has been shown that algorithms based on DR/ADMM often enjoy similar convergence properties also in the absence of strong convexity. For example, Liang et al. (2017) derived local linear convergence guarantees under mild assumptions; Applegate et al. (2021) established global linear convergence for Primal-Dual methods for LPs using restart strategies; and Wang & Shroff (2017) established linear convergence for an ADMM-based algorithm on general LPs. Yet, these frameworks quickly become intractable on large OT problems, due to the many memory operations required.

## 1.2 CONTRIBUTIONS

We demonstrate that DR splitting, when properly designed and implemented, can solve large-scale OT problems reliably and quickly, while retaining the excellent memory footprint and parallelization properties of the Sinkhorn method. Specifically, we make the following contributions:

- We develop a DR splitting algorithm that solves the original OT directly, avoiding the numerical issues of SK and forgoing the need for solving linear systems of general solvers. We perform simplifications to eliminate variables so that the final algorithm can be executed with only one matrix variable, while maintaining the same degree of parallelization. Our method implicitly maintains a primal-dual pair, which facilitates a simple evaluation of stopping criteria.

- We derive an iteration complexity $O(1/\epsilon)$ for our method. This is a significant improvement over the best-known estimate $O(1/\epsilon^2)$ for the Sinkhorn method (cf. Lin et al. (2019)). We also provide a global linear convergence rate that holds independently of the initialization, despite the absence of strong convexity in the OT problem.

- We detail an efficient GPU implementation that fuses many immediate steps into one and performs several on-the-fly reductions between a read and write of the matrix variable. We also show how a primal-dual stopping criterion can be evaluated at no extra cost. The implementation is

available as open source and gives practitioners a fast and robust OT solver also for applications where regularized OT is not suitable.

As a by-product of solving the original OT problem, our approximate solution is guaranteed to be sparse. Indeed, it is known that DR can identify an optimal support in a finite time, even before reaching a solution Iutzeler & Malick (2020). To avoid cluttering the presentation, we focus on the primal OT, but note that the approach applies equally well to the dual form. Moreover, our implementation is readily extended to other splitting schemes (e.g. Chambolle & Pock (2011)).

## 2 BACKGROUND

**Notation** For any $x, y \in \mathbb{R}^n$, $\langle x, y \rangle$ is the Euclidean inner product of $x$ and $y$, and $\|\cdot\|_2$ denotes the $\ell_2$-norm. For matrices $X, Y \in \mathbb{R}^{m \times n}$, $\langle X, Y \rangle = \text{tr}(X^\top Y)$ denotes their inner product and $\|\cdot\|_F = \sqrt{\langle \cdot, \cdot \rangle}$ is the the induced Frobenius norm. We use $\|\cdot\|$ to indicate either $\|\cdot\|_2$ or $\|\cdot\|_F$. For a closed and convex set $\mathcal{X}$, the distance and the projection map are given by $\text{dist}(x, \mathcal{X}) = \min_{z \in \mathcal{X}} \|z - x\|$ and $\mathsf{P}_{\mathcal{X}}(x) = \text{argmin}_{z \in \mathcal{X}} \|z - x\|$, respectively. Further, we denote the proximal operator of a closed convex function $f$ by $\text{prox}_{\rho f}(x) = \text{argmin}_z f(z) + \frac{1}{2\rho} \|z - x\|^2$, $\rho > 0$. The Euclidean projection of $x \in \mathbb{R}^n$ onto the nonnegative orthant is denoted by $[x]_+ = \max(x, 0)$, and $\Delta_n = \{x \in \mathbb{R}^n_+ : \sum_{i=1}^n x_i = 1\}$ is the $(n-1)$ dimensional probability simplex.

**OT and optimality conditions** Let $e = \mathbf{1}_n$ and $f = \mathbf{1}_m$, and consider the linear mapping

$$\mathcal{A} : \mathbb{R}^{m \times n} \to \mathbb{R}^{m+n} : X \mapsto (Xe, X^\top f),$$

and its adjoint $\mathcal{A}^* : \mathbb{R}^{m+n} \to \mathbb{R}^{m \times n} : (y, x) \mapsto ye^\top + fx^\top$. The projection onto the range of $\mathcal{A}$ is $\mathsf{P}_{\text{ran}\mathcal{A}}((y, x)) = (y, x) - \alpha(f, -e)$, where $\alpha = (f^\top y - e^\top x)/(m+n)$ (Bauschke et al., 2021). With $b = (p, q) \in \mathbb{R}^{m+n}$, Problem (1) can be written as a linear program on the form

$$\underset{X \in \mathbb{R}^{m \times n}}{\text{minimize}} \quad \langle C, X \rangle \quad \text{subject to} \quad \mathcal{A}(X) = b, \quad X \geq 0. \tag{2}$$

Let $(\mu, \nu) \in \mathbb{R}^{m+n}$ be the dual variable associated the affine constraint in (2). Then, using the definition of $\mathcal{A}^*$, the dual problem of (2), or equivalently of (1), reads

$$\underset{\mu, \nu}{\text{maximize}} \quad p^\top \mu + q^\top \nu \quad \text{subject to} \quad \mu e^\top + f\nu^\top \leq C. \tag{3}$$

OT is a bounded program and always admits a feasible solution. It is thus guaranteed to have an optimal solution, and the optimality conditions can be summarized as follows.

**Proposition 1.** *A pair $X$ and $(\mu, \nu)$ are primal and dual optimal if and only if: (i) $Xe = p$, $X^\top f = q$, $X \geq 0$, (ii) $\mu e^\top + f\nu^\top - C \leq 0$, (iii) $\langle X, C - \mu e^\top - f^\top \nu \rangle = 0$.*

These conditions mean that: (i) $X$ is primal feasible; (ii) $\mu, \nu$ are dual feasible, and (iii) the duality gap is zero $\langle C, X \rangle = p^\top \mu + q^\top \nu$, or equivalently, complementary slackness holds. Thus, solving the OT problem amounts to (approximately) finding such a primal-dual pair. To this end, we rely on the celebrated Douglas-Rachford splitting method (Lions & Mercier, 1979; Eckstein & Bertsekas, 1992; Fukushima, 1996; Bauschke & Combettes, 2017).

**Douglas-Rachford splitting** Consider composite convex optimization problems on the form

$$\underset{x \in \mathbb{R}^n}{\text{minimize}} \quad f(x) + g(x), \tag{4}$$

where $f$ and $g$ are proper closed and convex functions. To solve Problem (4), the DR splitting algorithm starts from $y_0 \in \mathbb{R}^n$ and repeats the following steps:

$$x_{k+1} = \text{prox}_{\rho f}(y_k), \quad z_{k+1} = \text{prox}_{\rho g}(2x_{k+1} - y_k), \quad y_{k+1} = y_k + z_{k+1} - x_{k+1}, \tag{5}$$

where $\rho > 0$ is a penalty parameter; $\text{prox}_{\rho f}(\cdot)$ and $\text{prox}_{\rho g}(\cdot)$ are the proximal operator of $f$ and $g$, respectively. Procedure (5) can be viewed as a fixed-point iteration $y_{k+1} = T(y_k)$ for the mapping

$$T(y) = y + \text{prox}_{\rho g}(2\text{prox}_{\rho f}(y) - y) - \text{prox}_{\rho f}(y). \tag{6}$$

The DR iterations (5) can also be derived from the ADMM method applied to an equivalent problem to (4) (see, Appendix C). Indeed, they are both special instances of the classical proximal point method in Rockafellar (1976). As for convergence, Lions & Mercier (1979) showed that $T$ is a *firmly-nonexpansive* mapping, from which they obtained convergence of $y_k$. Moreover, the sequence $x_k$ is guaranteed to converge to a minimizer of $f + g$ (assuming a minimizer exists) for any choice of $\rho > 0$. In particular, we have the following general convergence result, whose proof can be found in Bauschke & Combettes (2017, Corollary 28.3).

**Lemma 2.1.** *Consider the composite problem* (4) *and its Fenchel–Rockafellar dual defined as*

$$\underset{u \in \mathbb{R}^n}{\text{maximize}} \quad -f^*(-u) - g^*(u). \tag{7}$$

*Let $\mathcal{P}^\star$ and $\mathcal{D}^\star$ be the sets of solutions to the primal* (4) *and dual* (7) *problems, respectively. Let $x_k, y_k,$ and $z_k$ be generated by procedure* (5) *and let $u_k := (y_{k-1} - x_k)/\rho$. Then, there exists $y^\star \in \mathbb{R}^n$ such that $y_k \to y^\star$. Setting $x^\star = \text{prox}_{\rho f}(y^\star)$ and $u^\star = (y^\star - x^\star)/\rho$, then it holds that (i) $x^\star \in \mathcal{P}^\star$ and $u^\star \in \mathcal{D}^\star$; (ii) $x_k - z_k \to 0$, $x_k \to x^\star$ and $z_k \to x^\star$; (iii) $u_k \to u^\star$.*

## 3 DOUGLAS-RACHFORD SPLITTING FOR OPTIMAL TRANSPORT

To efficiently apply DR to OT, we need to specify the functions $f$ and $g$ as well as how to evaluate their proximal operations. We begin by introducing a result for computing the projection onto the set of *real-valued* matrices with prescribed row and column sums (Romero, 1990; Bauschke et al., 2021).

**Lemma 3.1.** *Let $e = \mathbf{1}_n$ and $f = \mathbf{1}_m$. Let $p \in \Delta_m$ and $q \in \Delta_n$. Then, the set $\mathcal{X}$ defined by:*

$$\mathcal{X} := \left\{ X \in \mathbb{R}^{m \times n} \,\middle|\, Xe = p \text{ and } X^\top f = q \right\}$$

*is non-empty, and for every given $X \in \mathbb{R}^{m \times n}$, we have*

$$\mathsf{P}_{\mathcal{X}}(X) = X - \frac{1}{n} \left( (Xe - p) e^\top - \gamma f e^\top \right) - \frac{1}{m} \left( f(X^\top f - q)^\top - \gamma f e^\top \right),$$

*where $\gamma = f^\top (Xe - p)/(m + n) = e^\top \left( X^\top f - q \right)/(m + n)$.*

The lemma follows immediately from Bauschke et al. (2021, Corollary 3.4) and the fact that $(p, q) = \mathsf{P}_{\text{ran}\mathcal{A}}((p, q))$. It implies that $\mathsf{P}_{\mathcal{X}}(X)$ can be carried out by basic linear algebra operations, such as matrix-vector multiplies and rank-one updates, that can be effectively parallelized.

### 3.1 ALGORITHM DERIVATION

Our algorithm is based on re-writing (2) on the standard form for DR-splitting (4) using a carefully selected $f$ and $g$ that ensures a rapid convergence of the iterates and an efficient execution of the iterations. In particular, we propose to select $f$ and $g$ as follows:

$$f(X) = \langle C, X \rangle + \mathsf{I}_{\mathbb{R}_+^{m \times n}}(X) \quad \text{and} \quad g(X) = \mathsf{I}_{\{Y : \mathcal{A}(Y) = b\}}(X). \tag{8}$$

By Lemma 3.1, we readily have the explicit formula for the proximal operator of $g$, namely, $\text{prox}_{\rho g}(\cdot) = \mathsf{P}_{\mathcal{X}}(\cdot)$. The proximal operator of $f$ can also be evaluated explicitly as:

$$\text{prox}_{\rho f}(X) = \mathsf{P}_{\mathbb{R}_+^{m \times n}}(X - \rho C) = [X - \rho C]_+.$$

The Douglas-Rachford splitting applied to this formulation of the OT problem then reads:

$$X_{k+1} = [Y_k - \rho C]_+, \quad Z_{k+1} = \mathsf{P}_{\mathcal{X}}(2X_{k+1} - Y_k), \quad Y_{k+1} = Y_k + Z_{k+1} - X_{k+1}. \tag{9}$$

Despite their apparent simplicity, the updates in (9) are inefficient to execute in practice due to the many *memory operations* needed to operate on the large arrays $X_k, Y_k, Z_k$ and $C$. To reduce the memory access, we will now perform several simplifications to eliminate variables from (9). The resulting algorithm can be executed with only one matrix variable while maintaining the same degree of parallelization.

We first note that the linearity of $\mathsf{P}_{\mathcal{X}}(\cdot)$ allows us to eliminate $Z$, yielding the $Y$-update

$$Y_{k+1} = X_{k+1} - n^{-1} \left( 2X_{k+1}e - Y_k e - p - \gamma_k f \right) e^\top - m^{-1} f \left( 2X_{k+1}^\top f - Y_k^\top f - q - \gamma_k e \right)^\top,$$

---

**Algorithm 1** Douglas-Rachford Splitting for Optimal Transport (DROT)

---

**Input:** OT($C, p, q$), initial point $X_0$, penalty parameter $\rho$
  1: $\phi_0 = 0, \varphi_0 = 0$
  2: $a_0 = X_0 e - p, b_0 = X_0^\top f - q, \alpha_0 = f^\top a_0/(m+n)$
  3: **for** $k = 0, 1, 2, \ldots$ **do**
  4:      $X_{k+1} = \left[X_k + \phi_k e^\top + f \varphi_k^\top - \rho C\right]_+$
  5:      $r_{k+1} = X_{k+1} e - p, s_{k+1} = X_{k+1}^\top f - q, \beta_{k+1} = f^\top r_{k+1}/(m+n)$
  6:      $\phi_{k+1} = (a_k - 2r_{k+1} + (2\beta_{k+1} - \alpha_k)f)/n$
  7:      $\varphi_{k+1} = (b_k - 2s_{k+1} + (2\beta_{k+1} - \alpha_k)e)/m$
  8:      $a_{k+1} = a_k - r_{k+1}, b_{k+1} = b_k - s_{k+1}, \alpha_{k+1} = \alpha_k - \beta_{k+1}$
  9: **end for**
**Output:** $X_K$

---

where $\gamma_k = f^\top \left(2X_{k+1}e - Y_k e - p\right)/(m+n) = e^\top \left(2X_{k+1}^\top f - Y_k^\top f - q\right)/(m+n)$. We also define the following quantities that capture how $Y_k$ affects the update of $Y_{k+1}$

$$a_k = Y_k e - p, \quad b_k = Y_k^\top f - q, \quad \alpha_k = f^\top a_k/(m+n) = e^\top b_k/(m+n).$$

Similarly, for $X_k$, we let:

$$r_k = X_k e - p, \quad s_k = X_k^\top f - q, \quad \beta_k = f^\top r_k/(m+n) = e^\top s_k/(m+n).$$

Recall that the pair $(r_k, s_k)$ represents the primal residual at $X_k$. Now, the preceding update can be written compactly as

$$Y_{k+1} = X_{k+1} + \phi_{k+1} e^\top + f \varphi_{k+1}^\top, \tag{10}$$

where

$$\phi_{k+1} = (a_k - 2r_{k+1} + (2\beta_{k+1} - \alpha_k)f)/n$$
$$\varphi_{k+1} = (b_k - 2s_{k+1} + (2\beta_{k+1} - \alpha_k)e)/m.$$

We can see that the $Y$-update can be implemented using 4 matrix-vector multiples (for computing $a_k, b_k, r_{k+1}, s_{k+1}$), followed by 2 rank-one updates. As a rank-one update requires a read from an input matrix and a write to an output matrix, it is typically twice as costly as a matrix-vector multiply (which only writes the output to a vector). Thus, it would involve 8 memory operations of large arrays, which is still significant.

Next, we show that the $Y$-update can be removed too. Noticing that updating $Y_{k+1}$ from $Y_k$ and $X_{k+1}$ does not need the *full* matrix $Y_k$, but only the ability to compute $a_k$ and $b_k$. This allows us to use a single *physical* memory array to represent both the sequences $X_k$ and $Y_k$. Suppose that we overwrite the matrix $X_{k+1}$ as:

$$X_{k+1} \leftarrow X_{k+1} + \phi_{k+1} e^\top + f \varphi_{k+1}^\top,$$

then after the two rank-one updates, the $X$-array holds the value of $Y_{k+1}$. We can access the actual $X$-value again in the next update, which now reads: $X_{k+2} \leftarrow \left[X_{k+1} - \rho C\right]_+$. It thus remains to show that $a_k$ and $b_k$ can be computed efficiently. By multiplying both sides of (10) by $e$ and subtracting the result from $p$, we obtain

$$Y_{k+1} e - p = X_{k+1} e - p + \phi_{k+1} e^\top e + (\varphi_{k+1}^\top e)f.$$

Since $e^\top e = n$ and $(b_k - 2s_{k+1})^\top e = (m+n)(\alpha_k - 2\beta_{k+1})$, it holds that $(\varphi_{k+1}^\top e)f = (\alpha_k - 2\beta_{k+1})f$. We also have $\phi_{k+1} e^\top e = a_k - 2r_{k+1} + (2\beta_{k+1} - \alpha_k)f$. Therefore, we end up with an extremely simple recursive form for updating $a_k$:

$$a_{k+1} = a_k - r_{k+1}.$$

Similarly, we have $b_{k+1} = b_k - s_{k+1}$ and $\alpha_{k+1} = \alpha_k - \beta_{k+1}$. In summary, the DROT method can be implemented with a single matrix variable as summarized in Algorithm 1.

**Stopping criteria** It is interesting to note that while DROT directly solves to the primal problem (1), it maintains a pair of vectors that *implicitly* plays the role of the dual variables $\mu$ and $\nu$ in the dual problem (3). To get a feel for this, we note that the optimality conditions in Proposition 1 are equivalent to the existence of a pair $X^\star$ and $(\mu^\star, \nu^\star)$ such that

$$(X^\star e, X^{\star\top} f) = (p, q) \quad \text{and} \quad X^\star = \left[X^\star + \mu^\star e^\top + f\nu^{\star\top} - C\right]_+.$$

Here, the later condition encodes the nonnegativity constraint, the dual feasibility, and the zero duality gap. The result follows by invoking Deutsch (2001, Theorem 5.6(ii)) with the convex cone $\mathcal{C} = \mathbb{R}_+^{m \times n}$, $y = X^\star \in \mathcal{C}$ and $z = \mu^\star e^\top + f\nu^{\star\top} - C \in \mathcal{C}^\circ$. Now, compared to Step 4 in DROT, the second condition above suggests that $\phi_k/\rho$ and $\varphi_k/\rho$ are such implicit variables. To see why this is indeed the case, let $U_k = (Y_{k-1} - X_k)/\rho$. Then it is easy to verify that

$$(Z_k - X_k)/\rho = (X_k - Y_{k-1} + \phi_k e^\top + f\varphi_k^\top)/\rho = -U_k + (\phi_k/\rho)e^\top + f(\varphi_k/\rho)^\top.$$

By Lemma 2.1, we have $Z_k - X_k \to 0$ and $U_k \to U^\star \in \mathbb{R}^{m \times n}$, by which it follows that

$$(\phi_k/\rho)e^\top + f(\varphi_k/\rho)^\top \to U^\star.$$

Finally, by evaluating the conjugate functions $f^*$ and $g^*$ in Lemma 2.1, it can be shown that $U^\star$ must have the form $\mu^\star e^\top + f\nu^{\star\top}$, where $\mu^\star \in \mathbb{R}^m$ and $\nu^\star \in \mathbb{R}^n$ satisfying $\mu^\star e^\top + f\nu^{\star\top} \leq C$; see Appendix D for details. With such primal-dual pairs at our disposal, we can now explicitly evaluate their distance to the set of solutions laid out in Proposition 1 by considering: $r_{\text{primal}} = (\|r_k\|^2 + \|s_k\|^2)^{1/2}$, $r_{\text{dual}} = \|[(\phi_k/\rho)e^\top + f(\varphi_k/\rho)^\top - C]_+\|$, and gap $= \left| \langle C, X_k \rangle - (p^\top \phi_k + \varphi_k^\top q)/\rho \right|$. As problem (1) is feasible and bounded, Lemma 2.1 and strong duality guarantees that all the three terms will converge to zero. Thus, we terminate DROT when these become smaller than some user-specified tolerances.

## 3.2 CONVERGENCE RATES

In this section, we state the main convergence results of the paper, namely a sublinear and a linear rate of convergence of the given splitting algorithm. In order to establish the sublinear rate, we need the following function:

$$V(X, Z, U) = f(X) + g(Z) + \langle U, Z - X \rangle,$$

which is defined for $X \in \mathbb{R}_+^{m \times n}$, $Z \in \mathcal{X}$ and $U \in \mathbb{R}^{m \times n}$. We can now state the first result.

**Theorem 1.** *Let $X_k, Y_k, Z_k$ be the generated by procedure (9). Then, for any $X \in \mathbb{R}_+^{m \times n}$, $Z \in \mathcal{X}$, and $Y \in \mathbb{R}^{m \times n}$, we have*

$$V(X_{k+1}, Z_{k+1}, (Y - Z)/\rho) - V(X, Z, (Y_k - X_{k+1})/\rho) \leq \left(\|Y_k - Y\|^2 - \|Y_{k+1} - Y\|^2\right)/(2\rho).$$

*Furthermore, let $Y^\star$ be a fixed-point of $T$ in (6) and let $X^\star$ be a solution of (1) defined from $Y^\star$ in the manner of Lemma 2.1. Then, it holds that*

$$\langle C, \bar{X}_k \rangle - \langle C, X^\star \rangle \leq \left(\|Y_0\|^2/(2\rho) + 2\|X^\star\| \|Y_0 - Y^\star\|/\rho\right)/k$$

$$\|\bar{Z}_k - \bar{X}_k\| = \|Y_k - Y_0\|/k \leq 2\|Y_0 - Y^\star\|/k,$$

*where $\bar{X}_k = \sum_{i=1}^k X_i/k$ and $\bar{Z}_k = \sum_{i=1}^k Z_i/k$.*

The theorem implies that one can compute a solution satisfying $\langle C, X \rangle - \langle C, X^\star \rangle \leq \epsilon$ in $O(1/\epsilon)$ iterations. This is a significant improvement over the best-known iteration complexity $O(1/\epsilon^2)$ of the Sinkhorn method (cf. Lin et al. (2019)). Note that the linearity of $\langle C, \cdot \rangle$ allows to update the *scalar* value $\langle C, \bar{X}_k \rangle$ recursively without ever needing to form the ergodic sequence $\bar{X}_k$. Yet, in terms of rate, this result is still conservative, as the next theorem shows.

**Theorem 2.** *Let $X_k$ and $Y_k$ be generated by (9). Let $\mathcal{G}^\star$ be the set of fixed points of $T$ in (6) and let $\mathcal{X}^\star$ be the set of primal solutions to (1). Then, $\{Y_k\}$ is bounded, $\|Y_k\| \leq M$ for all $k$, and*

$$\text{dist}(Y_k, \mathcal{G}^\star) \leq \text{dist}(Y_0, \mathcal{G}^\star) \times r^k \quad \text{and} \quad \text{dist}(X_k, \mathcal{X}^\star) \leq \text{dist}(Y_0, \mathcal{G}^\star) \times r^{k-1},$$

*where $r = c/\sqrt{c^2 + 1} < 1$, $c = \gamma(1 + \rho(\|e\| + \|f\|)) \geq 1$, and $\gamma = \theta_{\mathcal{S}^\star}(1 + \rho^{-1}(M + 1))$. Here, $\theta_{\mathcal{S}^\star} > 0$ is a problem-dependent constant characterized by the primal-dual solution sets only.*

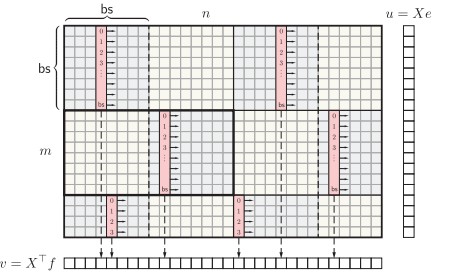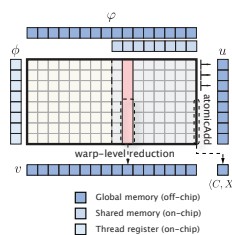

Figure 1: Left: Logical view of the main kernel. To expose sufficient parallelism to GPU, we organize threads into a 2D grid of blocks ($3 \times 2$ in the figure), which allows several threads per row. The threads are then grouped in 1D blocks (shown in red) along the columns of $X$. This ensures that global memory access is aligned and *coalesced* to maximize bandwidth utilization. We use the parameter work-size ws to indicate how many elements of a row each thread should handle. For simplicity, this parameter represents multiples of the block size bs. Each arrow denotes the activity of a single thread in a thread block. Memory storage is assumed to be column-major. Right: Activity of a normal working block, which handles a submatrix of size $\mathsf{bs} \times (\mathsf{ws} \cdot \mathsf{bs})$.

This means that an $\epsilon$-optimal solution can be computed in $O(\log 1/\epsilon)$ iterations. However, it is, in general, difficult to estimate the convergence factor $r$, and it may in the worst case be close to one. In such settings, the sublinear rate will typically dominate for the first iterations. In either case, DROT always satisfies the better of the two bounds at each $k$.

## 3.3 IMPLEMENTATION

In this section, we detail our implementation of DROT to exploit parallel processing on GPUs. We review only the most basic concepts of GPU programming necessary to describe our kernel and refer to Kirk & Wen-Mei (2016); Cheng et al. (2014) for comprehensive treatments.

**Thread hierarchy** When a kernel function is launched, a large number of threads are generated to execute its statements. These threads are organized into a two-level hierarchy. A *grid* contains multiple blocks and a *block* contains multiple threads. Each block is scheduled to one of the streaming multiprocessors (SMs) on the GPU concurrently or sequentially, depending on available hardware. While all threads in a thread block run *logically* in parallel, not all of them can run *physically* at the same time. As a result, different threads in a thread block may progress at a different pace. Once a thread block is scheduled to an SM, its threads are further partitioned into *warps*. A warp consists of 32 consecutive threads that execute the same instruction at *the same time*. Each thread has its own instruction address counter and register state, and carries out the current instruction on its own data.

**Memory hierarchy** *Registers* and *shared memory* ("on-chip") are the fastest memory spaces on a GPU. Registers are private to each thread, while shared memory is visible to all threads in the same thread block. An automatic variable declared in a kernel is generally stored in a register. Shared memory is programmable, and users have full control over when data is moved into or evicted from the shared memory. It enables block-level communication, facilitates reuse of on-chip data, and can greatly reduce the global memory access of a kernel. However, there are typically only a couple dozen registers per thread and a couple of kBs shared memory per thread block. The largest memory on a GPU card is *global memory*, physically separated from the compute chip ("off-chip"). All threads can access global memory, but its latency is much higher, typically hundreds of clock cycles.[1] Therefore, minimizing global memory transactions is vital to a high-performance kernel. When all threads of a *warp* execute a load (store) instruction that access *consecutive* memory locations, these will be *coalesced* into as few transactions as possible. For example, if they access consecutive 4-byte words such as *float32* values, four coalesced 32-byte transactions will service that memory access.

Before proceeding further, we state the main result of the section.

**Claim 3.1.** *On average, an iteration of DROT, including all the stopping criteria, can be done using 2.5 memory operations of $m \times n$ arrays. In particular, this includes one read from and write to $X$, and one read from $C$ in every other iteration.*

---

[1]https://docs.nvidia.com/cuda/cuda-c-programming-guide/index.html

**Main kernel**  Steps 4–5 and the stopping criteria are the main computational components of DROT, since they involve matrix operations. We will design an efficient kernel that: (i) updates $X_k$ to $X_{k+1}$, (ii) computes $u_{k+1} := X_{k+1}e$ and $v_{k+1} := X_{k+1}^\top f$, (iii) evaluates $\langle C, X_{k+1} \rangle$ in the duality gap expression. The kernel fuses many immediate steps into one and performs several on-the-fly reductions while updating $X_k$, thereby minimizing global memory access.

Our kernel is designed so that each thread block handles a *submatrix $x$* of size $\mathsf{bs} \times (\mathsf{ws} \cdot \mathsf{bs})$, except for the corner blocks which will have fewer rows and/or columns. Since all threads in a block need the same values of $\varphi$, it is best to read these into shared memory once per block and then let threads access them from there. We, therefore, divide $\varphi$ into chunks of the block size and set up a loop to let the threads collaborate in reading chunks in a coalesced fashion into shared memory. Since each thread works on a single row, it accesses the same element of $\phi$ throughout, and we can thus load and store that value directly in a register. These allow maximum reuse of $\varphi$ and $\phi$.

In the $j$-th step, the working block loads column $j$ of $x$ to the chip in a coalesced fashion. Each thread $i \in \{0, 1, \ldots, \mathsf{bs} - 1\}$ uses the loaded $x_{ij}$ to compute and store $x_{ij}^+$ in a register:

$$x_{ij}^+ = \max(x_{ij} + \phi_i + \varphi_j - \rho c_{ij}, 0),$$

where $c$ is the corresponding submatrix of $C$. As sum reduction is order-independent, it can be done locally at various levels, and local results can then be combined to produce the final value. We, therefore, reuse $x_{ij}^+$ and perform several such partial reductions. First, to compute the local value for $u_{k+1}$, at column $j$, thread $i$ simply adds $x_{ij}^+$ to a running sum kept in a register. The reduction leading to $\langle C, X_{k+1} \rangle$ can be done in the same way. The vertical reduction to compute $v_{k+1}$ is more challenging as coordination between threads is needed. We rely on *warp-level* reduction in which the data exchange is performed between registers. This way, we can also leverage efficient CUDA's built-in functions for collective communication at warp-level.[2] When done, the first thread in a warp holds the total reduced value and simply adds it atomically to a proper coordinate of $v_{k+1}$. Finally, all threads write $x_{ij}^+$ to the global memory. The process is repeated by moving to the next column.

When the block finishes processing the last column of the submatrix $x$, each thread adds its running sum atomically to a proper coordinate of $u_{k+1}$. They then perform one vertical (warp-level) reduction to collect their private sums to obtain the partial cost value $\langle c, x \rangle$. When all the submatrices have been processed, we are granted all the quantities described in (i)–(iii), as desired. It is essential to notice that each entry of $x$ is only read and written once during this process. To conclude Claim 3.1, we note that one can skip the load of $C$ in every other iteration. Indeed, if in iteration $k$, instead of writing $X_{k+1}$, one writes the value of $X_{k+1} - \rho C$, then the next update is simply $X_{k+2} = \left[ X_{k+1} + \phi_{k+1}e^\top + f\varphi_{k+1}^\top \right]_+$. Finally, all the remaining updates in DROT only involve vector and scalar operations, and can thus be finished off with a simple auxiliary kernel.

## 4  Experimental results

In this section, we perform experiments to validate our method and to demonstrate its efficiency both in terms of accuracy and speed. We focus on comparisons with the Sinkhorn method, as implemented in the POT toolbox[3], due to its minimal per-iteration cost and its publicly available GPU implementation. All runtime tests were carried out on an NVIDIA Tesla T4 GPU with 16GB of global memory. The CUDA C++ implementation of DROT is open source and available at https://github.com/vienmai/drot.

We consider six instances of SK and its log-domain variant, called SK1–SK6 and SK-Log-1–SK-Log-6, corresponding to $\eta = 10^{-4}, 10^{-3}, 5 \times 10^{-3}, 10^{-2}, 5 \times 10^{-2}, 10^{-1}$, in that order. Given $m$ and $n$, we generate source and target samples as $x_{\mathrm{s}}$ and $x_{\mathrm{t}}$, whose entries are drawn from a 2D Gaussian distribution with randomized means and covariances $(\mu_{\mathrm{s}}, \Sigma_{\mathrm{s}})$ and $(\mu_{\mathrm{t}}, \Sigma_{\mathrm{t}})$. Here, $\mu_{\mathrm{s}} \in \mathbb{R}^2$ has normal distributed entries, and $\Sigma_{\mathrm{s}} = A_{\mathrm{s}}A_{\mathrm{s}}^\top$, where $A_{\mathrm{s}} \in \mathbb{R}^{2 \times 2}$ is a matrix with random entries in $[0, 1]$; $\mu_{\mathrm{t}} \in \mathbb{R}^2$ has entries formed from $\mathcal{N}(5, \sigma_{\mathrm{t}})$ for $\sigma_{\mathrm{t}} > 0$, and $\Sigma_{\mathrm{t}} \in \mathbb{R}^{2 \times 2}$ is generated similarly to $\Sigma_{\mathrm{s}}$. Given $x_{\mathrm{s}}$ and $x_{\mathrm{t}}$, the cost matrix $C$ represents pair-wise squared distance between samples: $C_{ij} = \|x_{\mathrm{s}}[i] - x_{\mathrm{t}}[j]\|_2^2$ for $i \in \{1, \ldots, m\}$ and $j \in \{1, \ldots, n\}$ and is normalized to have $\|C\|_\infty = 1$. The marginals $p$ and $q$ are set to $p = \mathbf{1}_m/m$ and $q = \mathbf{1}_n/n$. DROT always starts at $X_0 = pq^\top$.

---

[2]https://developer.nvidia.com/blog/using-cuda-warp-level-primitives/
[3]https://pythonot.github.io

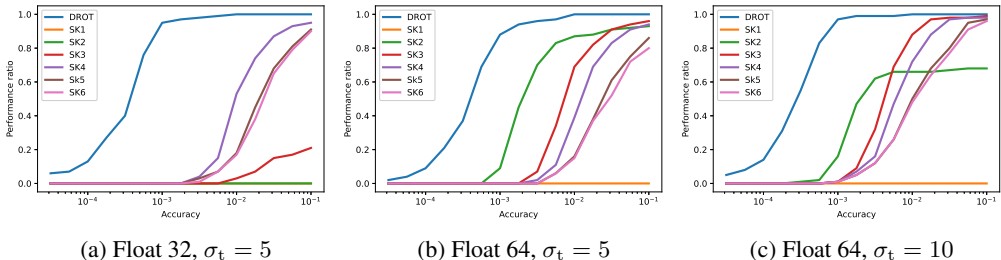

(a) Float 32, $\sigma_{\mathrm{t}} = 5$

(b) Float 64, $\sigma_{\mathrm{t}} = 5$

(c) Float 64, $\sigma_{\mathrm{t}} = 10$

Figure 2: The percentage of problems solved up to various accuracy levels for $\sigma_{\mathrm{t}} = 5, 10$.

**Performance profile**    For each method, we evaluate for each $\epsilon > 0$ the percentage of experiments for which $|f(X_K) - f(X^\star)| / f(X^\star) \leq \epsilon$, where $f(X^\star)$ is the optimal value and $f(X_K)$ is the objective value at termination. An algorithm is terminated as soon as the constraint violation goes below $10^{-4}$ or 1000 iterations have been executed. Figure 2 depicts the fraction of problems that are successfully solved up to an accuracy level $\epsilon$ given on the $x$-axis. Each subplot shows the result on 100 random problems with $m = n = 512$ and $\rho = 2/(m + n)$. We can see that DROT is consistently more accurate and robust. It reinforces that substantial care is needed to select the right $\eta$ for SK. The method is extremely vulnerable in single-precision and even in double-precision, an SK instance that seems to work in one setting can run into numerical issues in another. For example, by slightly changing a statistical properties of the underlying data, nearly $40\%$ of the problems in Fig. 2(c) cannot be solved by SK2 due to numerical errors, even though it works well in Fig. 2(b).

**Runtime**    As we strive for the excellent per-iteration cost of SK, this work would be incomplete if we do not compare these. Figure 3(a) shows the median of the per-iteration runtime and the 95% confidence interval, as a function of the dimensions. Here, $m$ and $n$ range from 100 to 20000 and each plot is obtained by performing 10 runs; in each run, the per-iteration runtime is averaged over 100 iterations. Since evaluating the termination criterion in SK is very expensive, we follow the POT default and only do so once every 10 iterations. The result confirms the efficiency of our kernel, and for very large problems the per-iteration runtimes of the two methods are almost identical. It is also clear that DROT and SK can execute an iteration much faster than the log-domain SK implementation. Finally, Figs 3(b)-(c) show the median times required for the total error (the sum of function gap and constraint violation) to reach different $\epsilon$ values. Since SK struggles to find even moderately accurate solutions in single precision, we drop these from the remaining runtime comparisons. Note also that by design, all methods start from different initial points.[4]  We can see that DROT can quickly and consistently find good approximates, while SK-Log is significantly slower for small $\eta$ and may not attain the desired accuracy with higher $\eta$.

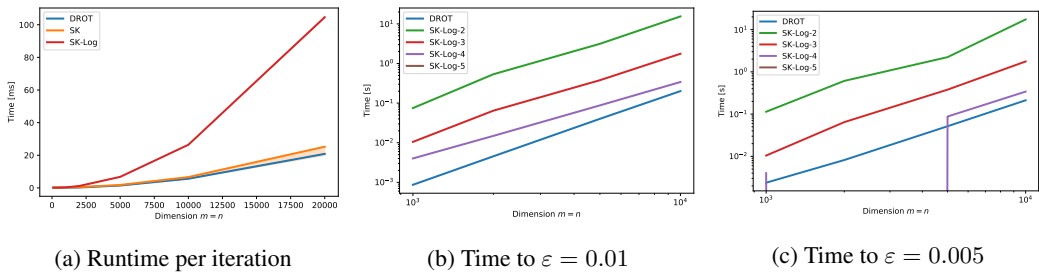

(a) Runtime per iteration

(b) Time to $\varepsilon = 0.01$

(c) Time to $\varepsilon = 0.005$

Figure 3: Wall-clock runtime performance versus the dimensions $m = n$ for $\sigma_{\mathrm{t}} = 5$.

The experiments confirm the strength of DROT both in speed and robustness: (i) each DROT iteration can be executed efficiently as in SK, and (ii) it can find high accuracy solutions as a well-tuned log-domain SK does, but at a much faster speed.

---

[4]The SK methods have their own matrices $K = \exp(-C/\eta)$ and $X_0 = \mathrm{diag}(u_0) K \, \mathrm{diag}(v_0)$. For DROT, since $X_{k+1} = [X_k + \phi_k e^\top f \varphi_k^\top - \rho C]_+$, where $X_0 = pq^\top = \mathbf{1}/(mn)$, $\|C\|_\infty = 1$, $\rho = \rho_0/(m + n)$, $\phi_0 = \varphi_0 = 0$, the term inside $[\cdot]_+$ is of the order $O(1/(mn) - \rho_0/(m + n)) \ll 0$. This makes the first few iterations of DROT identically zeros. To keep the canonical $X_0$, we just simply set $\rho_0$ to $0.5/\log(m)$ to reduce the warm-up period as $m$ increases, which is certainly suboptimal for DROT performance.

## ACKNOWLEDGEMENT

This work was supported in part by the Knut and Alice Wallenberg Foundation, the Swedish Research Council and the Swedish Foundation for Strategic Research, and the Wallenberg AI, Autonomous Systems and Software Program (WASP). The computations were enabled by resources provided by the Swedish National Infrastructure for Computing (SNIC), partially funded by the Swedish Research Council through grant agreement no. 2018-05973.

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

## A  Extra experiments

This section complements the experimental results in the main paper with further experiments for DROT, SK and SK-Log in various settings.

**Performance profile for MNIST**   We follow the construction in Cuturi (2013) and generate the cost matrix $C \in \mathbb{R}^{784 \times 784}$ as the normalized matrix of squared Euclidean distances between $28 \times 28$ bins in the grid. For each pair of source and target digits, we convert an image into a vector of intensity of $28 \times 28$ and normalize it to form the probability vectors $p$ and $q$, respectively. Similarly to Figure 2, Figure 4 shows the fraction of problems that are successfully solved up to an accuracy level $\varepsilon$ given on the x-axis. Here, each subplot is obtained by comparing 300 random pairs of images.

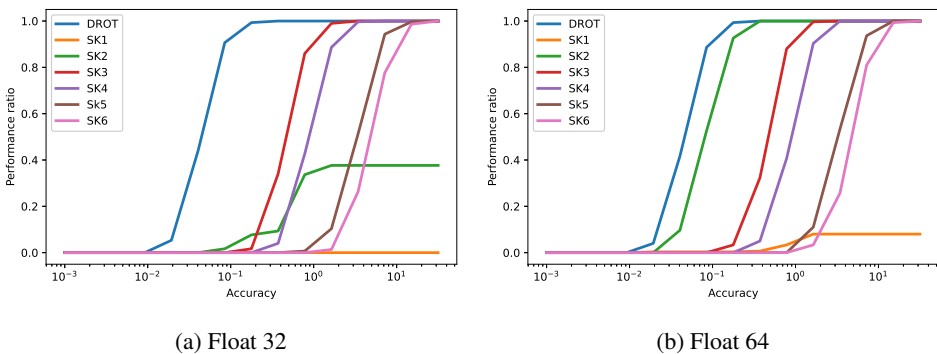

(a) Float 32  (b) Float 64

Figure 4: The percentage of problems solved up to various accuracy levels for MNIST.

**Performance profile for SK-Log**  Figure 5 shows the fraction of problems that are successfully solved by DROT and the log-domain implementation of SK. We can see that even with a simple stepsize rule, DROT achieves similar performance of a well-tuned log-domain SK method.

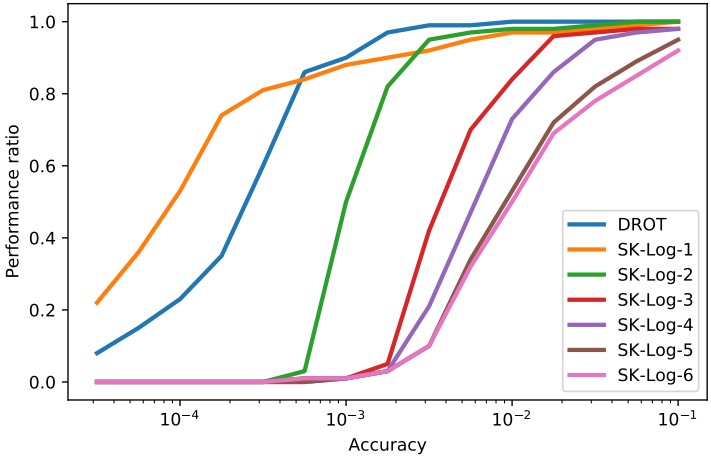

Figure 5: The percentage of problems solved up to various accuracy for $\sigma_{\mathrm{t}} = 5$ and $\rho = 2/(m+n)$.

**Single runs performance**  Figure 6 depicts the actual performance of different algorithms for random OT instances used to produce the performance profile in Figure 2. In addition, Figure 7 highlights the linear convergence phenomenon predicted by our theory.

**Sparsity of transportation**  By design, DROT efficiently finds sparse transport plans. To illustrate this, we apply DROT to a color transfer problem between two images (see Blondel et al. (2018)). By doing so, we obtain a highly sparse plan (approximately 99% zeros), and in turn, a high-quality

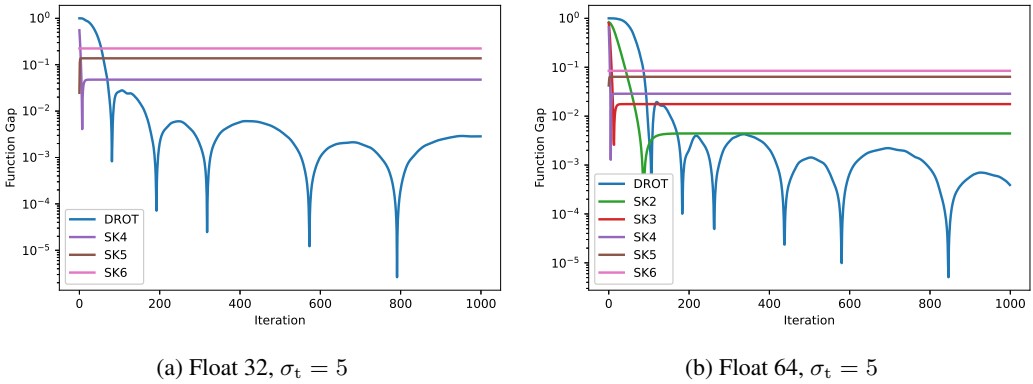

(a) Float 32, $\sigma_\mathrm{t} = 5$         (b) Float 64, $\sigma_\mathrm{t} = 5$

Figure 6: Objective suboptimality versus iteration for random OT instances with $m = n = 512$.

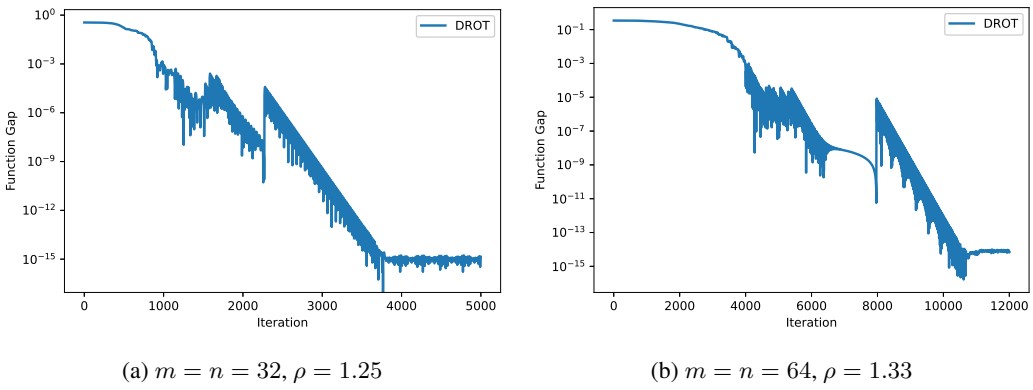

(a) $m = n = 32$, $\rho = 1.25$        (b) $m = n = 64$, $\rho = 1.33$

Figure 7: Objective suboptimality versus iteration for random OT instances with $\sigma_\mathrm{t} = 1$.

artificial image, visualized in Figure 8. In the experiment, we quantize each image with KMeans to reduce the number of distinct colors to 750. We subsequently use DROT to estimate an optimal color transfer between color distributions of the two images.

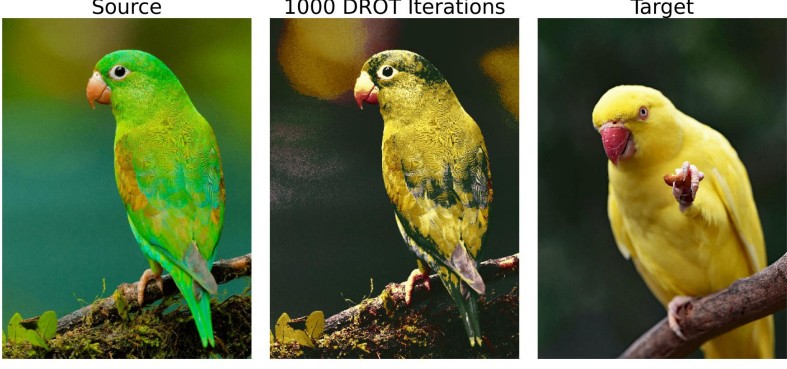

Figure 8: Color transfer via DROT: The left-most image is a KMeans compressed source image (750 centroids), the right-most is a compressed target image (also obtained via 750 KMeans centroids). The middle panel displays an artificial image generated by mapping the pixel values of each centroid in the source to a weighted mean of the target centroid. The weights are determined by the sparse transportation plan computed via DROT.

# B    PROOFS OF CONVERGENCE RATES

In order to establish the convergence rates for DROT, we first collect some useful results associated with the mapping that underpins the DR splitting:

$$T(y) = y + \text{prox}_{\rho g}\left(2\text{prox}_{\rho f}(y) - y\right) - \text{prox}_{\rho f}(y).$$

DR corresponds to finding a fixed point to $T$, i.e. a point $y^\star : y^\star = T(y^\star)$, or equivalently finding a point in the nullspace of $G$, where $G(y) = y - T(y)$. The operator $T$, and thereby also $G$, are *firmly non-expansive* (see e.g. Lions & Mercier (1979)). That is, for all $y, y' \in \text{dom}(T)$:

$$\langle T(y) - T(y'),\, y - y' \rangle \geq \|T(y) - T(y')\|^2$$
$$\langle G(y) - G(y'),\, y - y' \rangle \geq \|G(y) - G(y')\|^2. \tag{11}$$

Let $\mathcal{G}^\star$ be the set of all fixed points to $T$, i.e. $\mathcal{G}^\star = \{y \in \text{dom}(T)\,|\, G(y) = 0\}$. Then (11) implies the following bound:

**Lemma B.1.** *Let $y^\star \in \mathcal{G}^\star$ and $y_{k+1} = T(y_k)$, it holds that*

$$\|y_{k+1} - y^\star\|^2 \leq \|y_k - y^\star\|^2 - \|y_{k+1} - y_k\|^2.$$

*Proof.* We have

$$
\begin{aligned}
\|y_{k+1} - y^\star\|^2 &= \|y_{k+1} - y_k + y_k - y^\star\|^2 \\
&= \|y_k - y^\star\|^2 + 2\langle y_{k+1} - y_k, y_k - y^\star \rangle + \|y_k - y^\star\|^2 \\
&= \|y_k - y^\star\|^2 - 2\langle G(y_k) - G(y^\star), y_k - y^\star \rangle + \|y_k - y^\star\|^2 \\
&\leq \|y_k - y^\star\|^2 - 2\|G(y_k) - G(y^\star)\|^2 + \|y_k - y^\star\|^2 \\
&= \|y_k - y^\star\|^2 - \|y_{k+1} - y_k\|^2.
\end{aligned}
$$

$\square$

**Corollary B.1.** *Lemma B.1 implies that*

$$\|y_{k+1} - y_k\| \leq \|y_k - y^\star\| \tag{12a}$$
$$(\text{dist}(y_{k+1}, \mathcal{G}^\star))^2 \leq (\text{dist}(y_k, \mathcal{G}^\star))^2 - \|y_{k+1} - y_k\|^2. \tag{12b}$$

*Proof.* The first inequality follows directly from the non-negativity of the left-hand side of Lemma B.1. The latter inequality follows from:

$$(\text{dist}(y_{k+1}, \mathcal{G}^\star))^2 = \|y_{k+1} - \mathsf{P}_{\mathcal{G}^\star}(y_{k+1})\|^2 \leq \|y_{k+1} - \mathsf{P}_{\mathcal{G}^\star}(y_k)\|^2.$$

Applying Lemma B.1 to the right-hand side with $y^\star = \mathsf{P}_{\mathcal{G}^\star}(y_k)$ yields the desired result. $\square$

## B.1    PROOF OF THEOREM 1

Since DR is equivalent to ADMM up to changes of variables and swapping of the iteration order, it is expected that DR can attain the ergodic rate $O(1/k)$ of ADMM, both for the objective gap and the constraint violation (He & Yuan, 2012; Beck, 2017). However, mapping the convergence proof of ADMM to a specific instance of DR is tedious and error-prone. We thus give a direct proof here instead. We first recall the DROT method:

$$X_{k+1} = \left[Y_k - \rho C\right]_+ \tag{13a}$$
$$Z_{k+1} = \mathsf{P}_{\mathcal{X}}(2X_{k+1} - Y_k) \tag{13b}$$
$$Y_{k+1} = Y_k + Z_{k+1} - X_{k+1}. \tag{13c}$$

Since $X_{k+1}$ is the projection of $Y_k - \rho C$ onto $\mathbb{R}_+^{m \times n}$, it holds that

$$\langle X_{k+1} - Y_k + \rho C, X - X_{k+1} \rangle \geq 0 \quad \forall X \in \mathbb{R}_+^{m \times n}. \tag{14}$$

Also, since $\mathcal{X}$ is a closed affine subspace, we have

$$\langle Z_{k+1} - 2X_{k+1} + Y_k, Z - Z_{k+1} \rangle = 0 \quad \forall Z \in \mathcal{X}. \tag{15}$$

Next, for $X \in \mathbb{R}_+^{m \times n}$, $Z \in \mathcal{X}$ and $U \in \mathbb{R}^{m \times n}$ we define the function:

$$V(X, Z, U) = f(X) + g(Z) + \langle U, Z - X \rangle.$$

Let $U_{k+1} = (Y_k - X_{k+1})/\rho$, it holds that

$$\begin{aligned}
V&(X_{k+1}, Z_{k+1}, U_{k+1}) - V(X, Z, U_{k+1}) \\
&= f(X_{k+1}) + g(Z_{k+1}) + \langle U_{k+1}, Z_{k+1} - X_{k+1} \rangle - f(X) - g(Z) - \langle U_{k+1}, Z - X \rangle \\
&= \langle C, X_{k+1} \rangle - \langle C, X \rangle + \frac{1}{\rho} \langle Y_k - X_{k+1}, Z_{k+1} - Z + X - X_{k+1} \rangle.
\end{aligned}$$

By (14), we have $\langle Y_k - X_{k+1}, X - X_{k+1} \rangle / \rho \leq \langle C, X \rangle - \langle C, X_{k+1} \rangle$, we thus arrive at

$$\begin{aligned}
V&(X_{k+1}, Z_{k+1}, U_{k+1}) - V(X, Z, U_{k+1}) \leq \frac{1}{\rho} \langle Y_k - X_{k+1}, Z_{k+1} - Z \rangle \\
&= \frac{1}{\rho} \langle X_{k+1} - Z_{k+1}, Z_{k+1} - Z \rangle = -\frac{1}{\rho} \|X_{k+1} - Z_{k+1}\|^2 + \frac{1}{\rho} \langle X_{k+1} - Z_{k+1}, X_{k+1} - Z \rangle,
\end{aligned}$$
$$(16)$$

where the second step follows from (15). Now, for any $Y \in \mathbb{R}^{m \times n}$, we also have

$$\begin{aligned}
V&(X_{k+1}, Z_{k+1}, \frac{Y - Z}{\rho}) - V(X_{k+1}, Z_{k+1}, U_{k+1}) \\
&= \frac{1}{\rho} \langle Y - Z, Z_{k+1} - X_{k+1} \rangle - \frac{1}{\rho} \langle Y_k - X_{k+1}, Z_{k+1} - X_{k+1} \rangle \\
&= \frac{1}{\rho} \langle Y - Y_k, Z_{k+1} - X_{k+1} \rangle + \frac{1}{\rho} \langle X_{k+1} - Z, Z_{k+1} - X_{k+1} \rangle. \quad (17)
\end{aligned}$$

Therefore, by adding both sides of (16) and (17), we obtain

$$V(X_{k+1}, Z_{k+1}, \frac{Y - Z}{\rho}) - V(X, Z, U_{k+1}) \leq \frac{1}{\rho} \langle Y - Y_k, Z_{k+1} - X_{k+1} \rangle - \frac{1}{\rho} \|X_{k+1} - Z_{k+1}\|^2.$$
$$(18)$$

Since $Z_{k+1} - X_{k+1} = Y_{k+1} - Y_k$, it holds that

$$\frac{1}{\rho} \langle Y - Y_k, Z_{k+1} - X_{k+1} \rangle = \frac{1}{2\rho} \|Y_k - Y\|^2 - \frac{1}{2\rho} \|Y_{k+1} - Y\|^2 + \frac{1}{2\rho} \|Y_{k+1} - Y_k\|^2. \quad (19)$$

Plugging (19) into (18) yields

$$V(X_{k+1}, Z_{k+1}, \frac{Y - Z}{\rho}) - V(X, Z, U_{k+1}) \leq \frac{1}{2\rho} \|Y_k - Y\|^2 - \frac{1}{2\rho} \|Y_{k+1} - Y\|^2 - \frac{1}{2\rho} \|Y_{k+1} - Y_k\|^2,$$
$$(20)$$

which by dropping the last term on the right-hand side gives the first claim in the theorem.

Let $Y^\star$ be a fixed-point of the mapping $T$ and let $X^\star$ be a solution of (1) defined from $Y^\star$ in the manner of Lemma 2.1. Invoking (20) with $Z = X = X^\star \in \mathbb{R}_+^{m \times n} \cap \mathcal{X}$ and summing both sides of (20) over the iterations $0, \ldots, k-1$ gives

$$\langle C, \bar{X}_k \rangle - \langle C, X^\star \rangle + \left\langle \frac{Y - X^\star}{\rho}, \bar{Z}_k - \bar{X}_k \right\rangle \leq \frac{1}{k} \left( \frac{1}{2\rho} \|Y_0 - Y\|^2 - \frac{1}{2\rho} \|Y_k - Y\|^2 \right), \quad (21)$$

where $\bar{X}_k = \sum_{i=1}^k X_i / k$ and $\bar{Z}_k = \sum_{i=1}^k Z_i / k$. Since

$$\frac{1}{2\rho} \|Y_0 - Y\|^2 - \frac{1}{2\rho} \|Y_k - Y\|^2 = \frac{1}{2\rho} \|Y_0\|^2 - \frac{1}{2\rho} \|Y_k\|^2 + \frac{1}{\rho} \langle Y, Y_k - Y_0 \rangle,$$

it follows that

$$\langle C, \bar{X}_k \rangle - \langle C, X^\star \rangle + \frac{1}{\rho} \left\langle Y, \bar{Z}_k - \bar{X}_k - \frac{Y_k - Y_0}{k} \right\rangle \leq \frac{1}{2\rho k} \|Y_0\|^2 - \frac{1}{2\rho k} \|Y_k\|^2 + \frac{1}{\rho} \langle X^\star, \bar{Z}_k - \bar{X}_k \rangle.$$

Since $Y$ is arbitrary in the preceding inequality, we must have that

$$\bar{Z}_k - \bar{X}_k - \frac{Y_k - Y_0}{k} = 0, \tag{22}$$

from which we deduce that

$$
\begin{aligned}
\langle C, \bar{X}_k \rangle - \langle C, X^\star \rangle &\le \frac{1}{2\rho k} \|Y_0\|^2 - \frac{1}{2\rho k} \|Y_k\|^2 + \frac{1}{\rho} \langle X^\star, \bar{Z}_k - \bar{X}_k \rangle \\
&\le \frac{1}{2\rho k} \|Y_0\|^2 + \frac{1}{\rho} \|X^\star\| \|\bar{Z}_k - \bar{X}_k\| \\
&= \frac{1}{2\rho k} \|Y_0\|^2 + \frac{1}{\rho k} \|X^\star\| \|Y_k - Y_0\|.
\end{aligned}
\tag{23}
$$

By Lemma B.1, we have

$$\|Y_k - Y^\star\|^2 \le \|Y_{k-1} - Y^\star\|^2 \le \cdots \le \|Y_0 - Y^\star\|^2,$$

which implies that

$$\|Y_k - Y_0\| \le \|Y_k - Y^\star\| + \|Y_0 - Y^\star\| \le 2\|Y_0 - Y^\star\|. \tag{24}$$

Finally, plugging (24) into (23) and (22) yields

$$
\langle C, \bar{X}_k \rangle - \langle C, X^\star \rangle \le \frac{1}{k} \left( \frac{1}{2\rho} \|Y_0\|^2 + \frac{2}{\rho} \|X^\star\| \|Y_0 - Y^\star\| \right)
$$

$$
\|\bar{Z}_k - \bar{X}_k\| = \frac{\|Y_k - Y_0\|}{k} \le \frac{2\|Y_0 - Y^\star\|}{k},
$$

which concludes the proof.

### B.2 Proof of Theorem 2

We will employ a similar technique for proving linear convergence as Wang & Shroff (2017) used for ADMM on LPs. Although they, in contrast to our approach, handled the linear constraints via relaxation, we show that the linear convergence also holds for our OT-customized splitting algorithm which relies on polytope projections. The main difference is that the dual variables are given implicitly, rather than explicitly via the multipliers defined in the ADMM framework. As a consequence, their proof needs to be adjusted to apply to our proposed algorithm.

To facilitate the analysis of the given splitting algorithm, we let $e = \mathbf{1}_n$ and $f = \mathbf{1}_m$ and consider the following equivalent LP:

$$
\begin{aligned}
&\underset{X, Z \in \mathbb{R}^{m \times n}}{\text{minimize}} && \langle C, X \rangle \\
&\text{subject to} && Ze = p \\
& && Z^\top f = q \\
& && X \ge 0 \\
& && Z = X.
\end{aligned}
\tag{25}
$$

The optimality conditions can be written on the form

$$Z^\star e = p \tag{26a}$$

$$Z^{\star\top} f = q \tag{26b}$$

$$X^\star \ge 0 \tag{26c}$$

$$Z^\star = X^\star \tag{26d}$$

$$\mu^\star e^\top + f\nu^{\star\top} \le C \tag{26e}$$

$$\langle C, X^\star \rangle - p^\top \mu^\star - q^\top \nu^\star = 0. \tag{26f}$$

This means that set of optimal solutions, $\mathcal{S}^\star$, is a polyhedron

$$\mathcal{S}^\star = \{ \mathcal{Z} = (X, Z, \mu, \nu) \mid M_{\text{eq}}(\mathcal{Z}) = 0,\ M_{\text{in}}(\mathcal{Z}) \le 0 \}.$$

where $M_{\text{eq}}$ and $M_{\text{in}}$ denote the equality and inequality constraints of (26), respectively. Li (1994) proposed a uniform bound of the distance between a point such a polyhedron in terms of the constraint violation given by:

$$\text{dist}(\mathcal{Z}, \mathcal{S}^\star) \leq \theta_{\mathcal{S}^\star} \left\| \begin{array}{c} M_{\text{eq}}(\mathcal{Z}) \\ \left[ M_{\text{in}}(\mathcal{Z}) \right]_+ \end{array} \right\|. \tag{27}$$

For the $(X, Z)$-variables generated by (9), (26a), (26b), and (26c) will always hold due to the projections carried out in the subproblems. As a consequence, only (26d), (26e), and (26f) will contribute to the bound (27). In particular:

$$\text{dist}((X_k, Z_k, \mu_k, \nu_k), \mathcal{S}^\star) \leq \theta_{\mathcal{S}^\star} \left\| \begin{array}{c} Z_k - X_k \\ \langle C, X_k \rangle - p^\top \mu_k - q^\top \nu_k \\ \left[ \mu_k e^\top + f \nu_k^\top - C \right]_+ \end{array} \right\|. \tag{28}$$

The following Lemma allows us to express the right-hand side of (28) in terms of $Y_{k+1} - Y_k$.

**Lemma B.2.** *Let $X_k$, $Z_k$ and $Y_k$ be a sequence generated by* (9). *It then holds that*

$$Z_{k+1} - X_{k+1} = Y_{k+1} - Y_k$$
$$\mu_{k+1} e^\top + f \nu_{k+1}^\top - C \leq \rho^{-1}(Y_{k+1} - Y_k)$$
$$\langle C, X_{k+1} \rangle - p^\top \mu_{k+1} - q^\top \nu_{k+1} = -\rho^{-1} \langle Y_{k+1}, Y_{k+1} - Y_k \rangle.$$

*Proof.* The first equality follows directly from the $Y$-update in (9). Let

$$\mu_k = \phi_k/\rho, \quad \nu_k = \varphi_k/\rho \tag{29}$$

the $Y$-update in (10) yields

$$Y_{k+1} = Y_k + Z_{k+1} - X_{k+1} = X_{k+1} + \rho \mu_{k+1} e^\top + \rho f \nu_{k+1}^\top. \tag{30}$$

In addition, the $X$-update

$$X_{k+1} = \max(Y_k - \rho C, 0) \tag{31}$$

can be combined with the first equality in (30) to obtain

$$Y_{k+1} = \min(Y_k + Z_{k+1}, Z_{k+1} + \rho C),$$

or equivalently

$$Y_{k+1} - Z_{k+1} - \rho C = \min(Y_k - \rho C, 0). \tag{32}$$

By (30), the left-hand-side in (32) can be written as

$$Y_{k+1} - Z_{k+1} - \rho C = X_{k+1} - Z_{k+1} + \rho \mu_{k+1} e^\top + \rho f \nu_{k+1}^\top - \rho C$$
$$= \rho(\mu_{k+1} e^\top + f \nu_{k+1}^\top - C - \rho^{-1}(Y_{k+1} - Y_k)).$$

Substituting this in (32) gives

$$\mu_{k+1} e^\top + f \nu_{k+1}^\top - C - \rho^{-1}(Y_{k+1} - Y_k)) = \rho^{-1} \min(Y_k - \rho C, 0) \leq 0, \tag{33}$$

which gives the second relation in the lemma. Moreover, (31) and (33) implies that $X_{k+1}$ and $\mu_{k+1} e^\top + f \nu_{k+1}^\top - \rho^{-1}(Y_{k+1} - Y_k) - C$ cannot be nonzero simultaneously. As a consequence,

$$\left\langle X_{k+1}, \mu_{k+1} e^\top + f \nu_{k+1}^\top - \rho^{-1}(Y_{k+1} - Y_k) - C \right\rangle = 0$$

or

$$\begin{aligned} \langle C, X_{k+1} \rangle &= \left\langle X_{k+1}, \mu_{k+1} e^\top + f \nu_{k+1}^\top - \rho^{-1}(Y_{k+1} - Y_k) \right\rangle \\ &= \left\langle Z_{k+1} - (Y_{k+1} - Y_k), \mu_{k+1} e^\top + f \nu_{k+1}^\top - \rho^{-1}(Y_{k+1} - Y_k) \right\rangle \\ &= \left\langle Z_{k+1}, \mu_{k+1} e^\top + f \nu_{k+1}^\top \right\rangle - \left\langle \rho^{-1} Z_{k+1} + \mu_{k+1} e^\top + f \nu_{k+1}^\top, Y_{k+1} - Y_k \right\rangle \\ &\quad + \rho^{-1} \| Y_{k+1} - Y_k \|^2. \end{aligned}$$

By (30), $\rho^{-1}Z_{k+1} + \mu_{k+1}e^\top + f\nu_{k+1}^\top = \rho^{-1}(Z_{k+1} + Y_{k+1} - X_{k+1}) = \rho^{-1}(2Y_{k+1} - Y_k)$, hence

$$\langle C, X_{k+1}\rangle = \langle Z_{k+1}, \mu_{k+1}e^\top + f\nu_{k+1}^\top\rangle - \rho^{-1}\langle 2Y_{k+1} - Y_k, Y_{k+1} - Y_k\rangle + \rho^{-1}\|Y_{k+1} - Y_k\|^2$$
$$= \langle Z_{k+1}, \mu_{k+1}e^\top + f\nu_{k+1}^\top\rangle - \rho^{-1}\langle Y_{k+1}, Y_{k+1} - Y_k\rangle. \tag{34}$$

Note that

$$\langle Z_{k+1}, \mu_{k+1}e^\top + f\nu_{k+1}^\top\rangle = \mathrm{tr}(Z_{k+1}^\top \mu_{k+1}e^\top) + \mathrm{tr}(Z_{k+1}^\top f\nu_{k+1}^\top)$$
$$= \mathrm{tr}(e^\top Z_{k+1}^\top \mu_{k+1}) + \mathrm{tr}(\nu_{k+1}^\top Z_{k+1}^\top f)$$
$$= \mathrm{tr}(p^\top \mu_{k+1}) + \mathrm{tr}(\nu_{k+1}^\top q)$$
$$= p^\top \mu_{k+1} + q^\top \nu_{k+1}. \tag{35}$$

Substituting (35) into (34), we get the third equality of the lemma:

$$\langle C, X_{k+1}\rangle - p^\top \mu_{k+1} + q^\top \nu_{k+1} = -\rho^{-1}\langle Y_{k+1}, Y_{k+1} - Y_k\rangle.$$

$\square$

By combining Lemma B.2 and (28), we obtain the following result:

**Lemma B.3.** *Let $X_k$, $Z_k$ and $Y_k$ be a sequence generated by (9) and $\mu_k$, $\nu_k$ be defined by (29). Then, there is a $\gamma > 0$ such that:*

$$\mathrm{dist}((X_k, Z_k, \mu_k, \nu_k), \mathcal{S}^\star) \leq \gamma \|Y_k - Y_{k-1}\|.$$

*Proof.* Let $Y^\star \in \mathcal{G}^\star$, then Lemma B.1 asserts that

$$\|Y_k - Y^\star\| \leq \|Y_{k-1} - Y^\star\| \leq \cdots \leq \|Y_0 - Y^\star\|,$$

which implies that $\|Y_k\| \leq M$ for some positive constant $M$. In fact, $\{Y_k\}$ belongs to a ball centered at $Y^\star$ with the radius $\|Y_0 - Y^\star\|$. From the bound of (28) and Lemma B.2, we have

$$\mathrm{dist}((X_k, Z_k, \mu_k, \nu_k), \mathcal{S}^\star) \leq \theta_{\mathcal{S}^\star} \left\| \begin{matrix} Z_k - X_k \\ \langle C, X_k\rangle - p^\top \mu_k - q^\top \nu_k \\ [\mu_k e^\top + f\nu_k^\top - C]_+ \end{matrix} \right\|$$
$$\leq \theta_{\mathcal{S}^\star}(\|Z_k - X_k\| + |\langle C, X_k\rangle - p^\top \mu_k - q^\top \nu_k|$$
$$+ \|[\mu_k e^\top + f\nu_k^\top - C]_+\|)$$
$$\leq \theta_{\mathcal{S}^\star}(\|Y_k - Y_{k-1}\| + \rho^{-1}|\langle Y_k, Y_k - Y_{k-1}\rangle| + \rho^{-1}\|[Y_k - Y_{k-1}]_+\|)$$
$$\leq \theta_{\mathcal{S}^\star}(1 + \rho^{-1} + \rho^{-1}\|Y_k\|)\|Y_k - Y_{k-1}\|$$
$$\leq \theta_{\mathcal{S}^\star}(1 + \rho^{-1}(M + 1))\|Y_k - Y_{k-1}\|.$$

By letting $\gamma = \theta_{\mathcal{S}^\star}(1 + \rho^{-1}(M + 1))$ we obtain the desired result. $\square$

Recall that for DROT, $\mathrm{prox}_{\rho f}(\cdot) = \mathsf{P}_{\mathbb{R}_+^{m \times n}}(\cdot - \rho C)$ and $\mathrm{prox}_{\rho g}(\cdot) = \mathsf{P}_{\mathcal{X}}(\cdot)$. The following Lemma bridges the optimality conditions stated in (26) with $\mathcal{G}^\star$.

**Lemma B.4.** *If $Y^\star = X^\star + \rho(\mu^\star e^\top + f\nu^{\star\top})$, where $X^\star$, $\mu^\star$, and $\nu^\star$ satisfy (26), then $G(Y^\star) = 0$.*

*Proof.* We need to prove that

$$G(Y^\star) = \mathsf{P}_{\mathbb{R}_+^{m \times n}}(Y^\star - \rho C) - \mathsf{P}_{\mathcal{X}}\left(2\mathsf{P}_{\mathbb{R}_+^{m \times n}}(Y^\star - \rho C) - Y^\star\right)$$

is equal to zero. Let us start with simplifying $\mathsf{P}_{\mathbb{R}_+^{m \times n}}(Y^\star - \rho C)$. By complementary slackness,

$$\langle X^\star, \mu^\star e^\top + f\nu^{\star\top} - C\rangle = 0,$$

and by dual feasibility

$$\mu^\star e^\top + f\nu^{\star\top} - C \leq 0.$$

Consequentially, we have:

$$\langle \mu^\star e^\top + f\nu^{\star\top} - C, X - X^\star \rangle \leq 0, \quad \forall X \in \mathbb{R}_+^{m \times n},$$

which defines a normal cone of $\mathbb{R}_+^{m \times n}$, i.e. $\mu^\star e^\top + f\nu^{\star\top} - C \in N_{\mathbb{R}_+^{m \times n}}(X^\star)$. By using the definition of $Y^\star$, and that $\rho > 0$, this implies that

$$Y^\star - X^\star - \rho C \in N_{\mathbb{R}_+^{m \times n}}(X^\star),$$

which corresponds to optimality condition of the projection

$$X^\star = \mathsf{P}_{\mathbb{R}_+^{m \times n}}(Y^\star - \rho C).$$

We thus have $\mathsf{P}_{\mathcal{X}}\left(2\mathsf{P}_{\mathbb{R}_+^{m \times n}}(Y^\star - \rho C) - Y^\star\right) = \mathsf{P}_{\mathcal{X}}(2X^\star - Y^\star)$. But by the definition of $Y^\star$,

$$0 = Y^\star - X^\star - \rho(\mu^\star e^\top + f\nu^{\star\top}) = X^\star - (2X^\star - Y^\star) - \rho\mu^\star e^\top - \rho f\nu^{\star\top},$$

which means that $X^\star = \mathsf{P}_{\mathcal{X}}(2X^\star - Y^\star)$ (following from the optimality condition of the projection). Substituting all these quantities into the formula for $G(Y^\star)$, we obtain $G(Y^\star) = 0$. $\qquad\square$

**Lemma B.5.** *There exists a constant $c \geq 1$ such that*

$$\mathrm{dist}(Y_k, \mathcal{G}^\star) \leq c \, \|Y_k - Y_{k-1}\|.$$

*Proof.* Let $Y^\star = X^\star + \rho\mu^\star e^\top + \rho f\nu^{\star\top}$, where $(X^\star, X^\star, \mu^\star, \nu^\star) = \mathsf{P}_{\mathcal{S}^\star}((X_k, Z_k, \mu_k, \nu_k))$. According to Lemma B.4, we have that $Y^\star \in G^\star$. Then

$$\mathrm{dist}(Y_k, \mathcal{G}^\star) \leq \|Y_k - Y^\star\| = \|Y_k - X^\star - \rho(\mu^\star e^\top + f\nu^{\star\top})\|.$$

Since $Y_k = X_k + \rho\mu_k e^\top + \rho f\nu_k^\top$, it holds that

$$\begin{aligned}
\mathrm{dist}(Y_k, \mathcal{G}^\star) &\leq \|X_k - X^\star + \rho(\mu_k - \mu^\star)e^\top + \rho f(\nu_k - \nu^\star)^\top\| \\
&\leq \|X_k - X^\star\| + \rho\|e\|\|\mu_k - \mu^\star\| + \rho\|f\|\|\nu_k - \nu^\star\| \\
&\leq \sqrt{1 + \rho^2\|e\|^2 + \rho^2\|f\|^2}\sqrt{\|X_k - X^\star\|^2 + \|\mu_k - \mu^\star\|^2 + \|\nu_k - \nu^\star\|^2} \\
&\leq \sqrt{1 + \rho^2\|e\|^2 + \rho^2\|f\|^2}\sqrt{\|(X_k - X^\star, Z_k - X^\star, \mu_k - \mu^\star, \nu_k - \nu^\star)\|^2} \\
&= \sqrt{1 + \rho^2\|e\|^2 + \rho^2\|f\|^2} \, \mathrm{dist}((X_k, Z_k, \mu_k, \nu_k), \mathcal{S}^\star) \\
&\leq \gamma(1 + \rho(\|e\| + \|f\|))\|Y_k - Y_{k-1}\|,
\end{aligned}$$

where the last inequality follows from Lemma B.3. The fact that

$$c = \gamma(1 + \rho(\|e\| + \|f\|)) \geq 1 \tag{36}$$

follows from (12a) in Corollary B.1. $\qquad\square$

### B.3 PROOF OF LINEAR CONVERGENCE

By combining Lemma B.5 with (12b) in Corollary B.1, we obtain:

$$(\mathrm{dist}(Y_{k+1}, \mathcal{G}^\star))^2 \leq \frac{c^2}{1 + c^2}(\mathrm{dist}(Y_k, \mathcal{G}^\star))^2$$

or equivalently

$$\mathrm{dist}(Y_{k+1}, \mathcal{G}^\star) \leq \frac{c}{\sqrt{1 + c^2}} \, \mathrm{dist}(Y_k, \mathcal{G}^\star).$$

Letting $r = c/\sqrt{1 + c^2} < 1$ and iterating the inequality $k$ times gives

$$\mathrm{dist}(Y_k, \mathcal{G}^\star) \leq \mathrm{dist}(Y_0, \mathcal{G}^\star)r^k.$$

Let $Y^\star := \mathsf{P}_{\mathcal{G}^\star}(Y_k)$. Since $\mathrm{prox}_{\rho f}(Y^\star) \in \mathcal{X}^\star$ and $X_{k+1} = \mathrm{prox}_{\rho f}(Y_k)$, we have

$$\mathrm{dist}(X_{k+1}, \mathcal{X}^\star) \leq \|\mathrm{prox}_{\rho f}(Y_k) - \mathrm{prox}_{\rho f}(Y^\star)\| \leq \|Y_k - Y^\star\| = \mathrm{dist}(Y_k, \mathcal{G}^\star) \leq \mathrm{dist}(Y_0, \mathcal{G}^\star)r^k,$$

where the second inequality follows from the non-expansiveness of $\mathrm{prox}_{\rho f}(\cdot)$ This completes the proof of Theorem 2.

## C  DR AND ITS CONNECTION TO ADMM

The Douglas-Rachford updates given in (5) can equivalently be formulated as:

$$x_{k+1} = \text{prox}_{\rho f}(y_k)$$
$$z_{k+1} = \text{prox}_{\rho g}(2x_{k+1} + y_k)$$
$$y_{k+1} = y_k + z_{k+1} - x_{k+1}$$
$$\Leftrightarrow$$
$$z_{k+1} = \text{prox}_{\rho g}(2x_{k+1} + y_k)$$
$$y_{k+1} = y_k + z_{k+1} - x_{k+1}$$
$$x_{k+2} = \text{prox}_{\rho f}(y_{k+1})$$
$$\Leftrightarrow$$
$$z_{k+1} = \text{prox}_{\rho g}(2x_{k+1} + y_k)$$
$$x_{k+2} = \text{prox}_{\rho f}(y_k + z_{k+1} - x_{k+1})$$
$$y_{k+1} = y_k + z_{k+1} - x_{k+1}.$$

If we let $w_{k+1} = \frac{1}{\rho}(y_k - x_{k+1})$, we get.

$$z_{k+1} = \text{prox}_{\rho g}(x_{k+1} + \rho w_{k+1})$$
$$x_{k+2} = \text{prox}_{\rho f}(z_{k+1} - \rho w_{k+1})$$
$$w_{k+2} = w_{k+1} + \frac{1}{\rho}(z_{k+1} - x_{k+2}).$$

By re-indexing the $x$ and the $w$ variables, we get:

$$z_{k+1} = \text{prox}_{\rho g}(x_k + \rho w_k)$$
$$x_{k+1} = \text{prox}_{\rho f}(z_{k+1} - \rho w_k)$$
$$w_{k+1} = w_k + \frac{1}{\rho}(z_{k+1} - x_{k+1}).$$

This is exactly ADMM applied to the composite convex problem on the form (4).

## D  FENCHEL-ROCKAFELLAR DUALITY OF OT

Recall that

$$f(X) = \langle C, X \rangle + I_{\mathbb{R}_+^{m \times n}}(X) \quad \text{and} \quad g(X) = I_{\{Y:\mathcal{A}(Y)=b\}}(X) = I_{\mathcal{X}}(X).$$

Recall also that the conjugate of the indicator function $I_\mathcal{C}$ of a closed and convex set $\mathcal{C}$ is the support function $\sigma_\mathcal{C}$ of the same set (Beck, 2017, Chapter 4). For a non-empty affine set $\mathcal{X} = \{Y : \mathcal{A}(Y) = b\}$, its support function can be computed as:

$$\sigma_\mathcal{X}(U) = \langle U, X_0 \rangle + I_{\text{ran}\mathcal{A}^*}(U),$$

where $X_0$ is a point in $\mathcal{X}$ (Beck, 2017, Example 2.30). By letting $X_0 = pq^\top \in \mathcal{X}$, it follows that

$$g^*(U) = \langle U, pq^\top \rangle + I_{\text{ran}\mathcal{A}^*}(U).$$

Since $\mathcal{A}^* : \mathbb{R}^{m+n} \to \mathbb{R}^{m \times n} : (y, x) \mapsto ye^\top + fx^\top$, any matrix $U \in \text{ran}\mathcal{A}^*$ must have the form $\mu e^\top + f\nu^\top$ for some $\mu \in \mathbb{R}^m$ and $\nu \in \mathbb{R}^n$. The function $g^*(U)$ can thus be evaluated as:

$$g^*(U) = p^\top \mu + q^\top \nu.$$

We also have

$$f^*(-U) = \max_X \{\langle -U, X \rangle - \langle C, X \rangle - I_{\mathbb{R}_+^{m \times n}}(X)\}$$
$$= \max_X \{\langle -U - C, X \rangle - I_{\mathbb{R}_+^{m \times n}}(X)\}$$
$$= \sigma_{\mathbb{R}_+^{m \times n}}(-U - C)$$
$$= \begin{cases} 0 & \text{if } -U - C \leq 0, \\ +\infty & \text{otherwise.} \end{cases}$$

The Fenchel–Rockafellar dual of problem 4 in Lemma 2.1 thus becomes

$$\begin{array}{ll} \underset{\mu \in \mathbb{R}^m, \nu \in \mathbb{R}^n}{\text{maximize}} & -p^\top \mu - q^\top \nu \\ \text{subject to} & -\mu e^\top - f\nu^\top - C \leq 0, \end{array}$$

which is identical to (3) upon replacing $\mu$ by $-\mu$ and $\nu$ by $-\nu$.

