# OpenReview forum: "A fast and accurate splitting method for optimal transport: analysis and implementation"
_ICLR.cc/2022/Conference — ICLR 2022 Poster_

### Official Review · Reviewer_szP9 · 2021-10-19

**Correctness:** 4
**Technical Novelty And Significance:** 3
**Empirical Novelty And Significance:** 3
**Recommendation:** 6
**Confidence:** 3

**Main Review:**

While alternate minimization type approaches for OT have been extensively studied, I find the performance of about 1 second for 20000-by-20000 instances highly impressive. This requires a fairly high degree of parallelism, and is quite difficulty to obtain without significantly adapting to hardware. Some of the subtleties, such as numerical instability of sinkhorn2, is also something that I have observed before, so I consider the experiments quite thorough.

The algorithms are theoretically well founded, and have convergence guarantees typical of this type of methods. They also have the advantage of working directly with regularized OT instances, which to my understanding are critical for the downstream applications using OT. My main worry about these theoretical guarantees is that they are quite difficult to compare vs. unconditional bounds such as those obtained via combinatorial optimization packages, or the recent (purely theoretical) n^2 type methods (https://arxiv.org/pdf/2101.05719.pdf).

**Summary Of The Paper:**

This paper studies a splitting based gradient descent method for computing optimal transport objectives. It's based on splitting the row/column sum constraints into two objectives, and using a variant of the DR splitting algorithm. The paper rigorously proves a convergence guarantee of 1/k times the initial distance after k steps. Then it explores ways of setting up its algorithm in ways friendly to parallelism / memory hierarchy empirically: they obtain a very good wall-clock performance of about 1 second on a 20,000-by-20,000 optimal transport instance, and also examine the convergence of these algorithms.

**Summary Of The Review:**

Overall, I believe this paper has useful ideas for improving the performance of OT methods, especially those currently in use, but it falls a bit short of proposing something that significantly advances the state-of-art for OT.

---

> ### Author Response · Authors · 2021-11-14
> **Response to Reviewer szP9**
>
>
> We thank the reviewer for appreciating our effort. We also thank the reviewer for summarising our work in both qualitative and quantitative terms.
>
> 1) **The reviewer concerns the difficulty in comparing the convergence guarantees with bounds achieved by combinatorial optimization packages.**
>
> We completely agree with the reviewer. However, we should stress that this is **not specific to our method** but rather the nature of the two different domains: continuous and combinatorial optimization. We try our best to summary some known results below:
>
> - To our knowledge, when applied to **real-valued** OTs, the best known complexity for combinatorial algorithms is $\tilde{O}(n^{2.5}/\log(\epsilon))$ (cf. [LHJ19, PC19]). This bound is obtained by an interior-point method using the Lee-Sidford barrier function. Recall that the time complexities for our method and Sinkhorn are $O(n^2/\epsilon)$ and $O(n^2/\epsilon^2)$, respectively. The improved rate in the new paper the reviewer pointed to seems to focus on minimum cost flow problems with **integer polynomially-bounded costs and capacities**. To be fully honest, we do not know how to connect this with OT problems.
>
>
> - On the practical side, while the Lee-Sidford method, introduced 7 years ago, is intellectually interesting, its implementation is still not yet available in the literature (cf. Section 1 in [LHJ19], as of July 2021). In addition, parallelization is crucial for actual performance on large OT instances. Unfortunately, combinatorial algorithms are known to be notoriously difficult to parallelize on GPUs due to conditional branchings. We feel that without such an actual implementation to compare with, it is really hard to make any comprehensive conclusions here.
>
> Nonetheless, the aforementioned ambiguities won't change the qualitative aspect of our contributions in any way.
>
> [LHJ19] T. Lin, N. Ho, and M. Jordan. On efficient optimal transport: An analysis of greedy and accelerated mirror descent algorithms, https://arxiv.org/pdf/1901.06482.pdf
>
> [PC19] G. Peyré and M. Cuturi. Computational optimal transport. *Foundations and Trends in Machine Learning*, 11(5-6):355–607, 2019.

---

> > ### Comment · Reviewer_szP9 · 2021-11-29
> > **approximations allow one to assume poly-bounded costs/capacities**
> >
> > re your point 1:
> >
> > at a high level, the recent methods are variants of LS optimized for graphs, so the same invocations work.
> >
> > slightly more detailed (for unit, instead of general demands): find the `bottleneck cost', remove all costs more than poly(n) * that cost, and set add 1 / poly(n) * that to all costs. Section 2.2 of https://arxiv.org/pdf/1010.2921.pdf has an example of this for approximate max-flows.
> >
> > also, I believe for practically relevant instances, it's reasonable to assume poly bounded costs/demands theoretically. For practical implementations, the above rounding is quite important (for avoiding logW, and only paying log(1/\eps))
> >
> > re point 2: I agree that LS is not the right algorithm to implement. However, my understanding is that on explicitly given (as adjacency matrix, as opposed to geometric) OT instances, standard combinatorial optimization packages do very well. I agree with the authors that this is an issue that this is a much broader issue, and does not need to be fully addressed here.

---

### Official Review · Reviewer_6N6U · 2021-10-31

**Correctness:** 4
**Technical Novelty And Significance:** 3
**Empirical Novelty And Significance:** 2
**Recommendation:** 6
**Confidence:** 3

**Main Review:**

This paper report a new efficient numerical scheme for the fundamental discrete OT problem. The main technical contribution is a new splitting scheme by a recent result on efficient range space projection, which is very interesting. The authors also parallelize the algorithm for GPU computing and made the procedure numerical stable and efficient. The theoretical results are somehow standard as DR splitting is a special case of ADMM, of which the standard convergence behavior is well understood now. The linear convergence is mainly due to the KKT mapping being polyhedral and thus metric subregular. Existing results could directly give the linear convergence.

My main questions are as follows:

* The experiments are mostly conducted compared with the Sinkhorn method. It would be interesting to see the results of other splitting schemes for discrete OT, e.g., the iterative Bregman projection.

* It would be good to also provide empirical objective function value convergence curves. As the global linear convergence (due to global Hoffman bound) is proved, it would be interesting to see the empirical performance for that.

* Could the 1/k convergence be made non-ergodic?


**Summary Of The Paper:**

This paper introduces a new splitting scheme for the original discrete OT problem. Unlike the existing splitting techniques, their solution requires an efficient implementation of rectangular matrix range space projection. The authors provide (somehow standard) 1/T convergence complexity and global linear convergence complexity. They report numerical results compared with the Sinkhorn method.

**Summary Of The Review:**

This paper introduces a new efficient DR scheme for discrete OT. The problem is important as it's widely used in ML models. The new splitting Eq.(7) is interesting, and the authors have spent an extensive effort to make the implementation efficient and robust.

---

> ### Author Response · Authors · 2021-11-14
> **Response to Reviewer 6N6U**
>
> We thank the reviewer for appreciating our effort and for such an encouraging evaluation of our work.
>
> 1. **The reviewer suggests comparing with the Iterative Bregman projection method**
>
> We thank the reviewer for bring our attention to the iterative Bregman projection method (IBP). Interestingly, we just learned that IBP applied to the two convex sets:
> $$
>     \mathcal{A} := \\{X : X\mathbf{1}_m = p\\} \quad \text{and} \quad
>     \mathcal{B} := \\{X : X^\top\mathbf{1}_n = q\\}
> $$
> is indeed **equivalent** to the Sinkhorn method, as shown in [PC19, Remark 4.8]. They also explicitly pointed out that one should prefer the original form of SK for more efficient executions in practice. We also refer to our response to Reviewer **qeU8** for more discussions on our choices of experiments.
>
> 2. **The reviewer suggests adding empirical objective value convergence curves.**
>
> Thank you for the suggestion. Due to the limited space, we have added in Appendix A a set of plots on the objective value versus the iteration count. The linear convergence phenomenon due to global Hoffman bound is reported in Figure 6. If we extrapolated the convergence slope to fully cover the $x$-axis, we would recognize that the sublinear rate will dominate for small $k$. Linear convergence becomes particularly useful for applications that need highly accurate solutions, say $\epsilon \leq 10^{-6}$. This reinforces the great adaptivity property of Douglas-Rachford splitting in that the method always satisfies the better of the two bounds at each $k$. We have updated the discussions after Theorems 1 and 2 to better stress this in the paper.
>
> 3. **Could the $O(1/k)$ convergence rate be made non-ergodic?**
>
> It is indeed possible. In particular, using a clever modificataion of Largrangian-based methods (including DR and ADMM as special cases) introduced in [ST20], the non-ergodic rate $O(1/k)$ can be achieved without further assumptions. However, its efficacy remains to be tested as their method requires extra matrix variables and matrix-vector multiplications. It should be noted that when computing the OT distance, the linearity of the objective allows to update the **scalar** $\langle C, \bar{X}_k \rangle$ recursively without ever needing to form the ergodic sequence.
>
> **References**
>
> [PC19] G. Peyré and M. Cuturi. Computational optimal transport. *Foundations and Trends in Machine Learning*, 11(5-6):355–607, 2019.
>
> [ST20] S. Sabach, M. Teboulle, Faster Lagrangian-based methods in convex optimization, arXiv:2010.14314.

---

### Official Review · Reviewer_qeU8 · 2021-11-01

**Correctness:** 3
**Technical Novelty And Significance:** 3
**Empirical Novelty And Significance:** Not applicable
**Recommendation:** 6
**Confidence:** 4

**Main Review:**

- It's a novel idea to rewrite the original OT problem into the standard form for DR-splitting and exploit the DR-splitting algorithm to solve it. The DROT method performs well in aspects of both speed and accuracy, thus being a strong competitor of the popular Sinkhorn method.

- The proposed algorithm can save memory efficiently for it can be executed with only a single matrix variable.

- The organization of this paper is satisfactory, and the flow of algorithm derivation is very clear.

- Experiments on much larger-scale datasets are needed to verify the effectiveness of the proposed OT solver.

- Log-domain stabilization of Sinkhorn proposed by [1] is also a popular strategy to deal with numerical issues introduced by entropy regularization, while there is a lack of comparison with it.

[1] Schmitzer, Bernhard. "Stabilized sparse scaling algorithms for entropy regularized transport problems." SIAM Journal on Scientific Computing 41.3 (2019): A1443-A1481.

There are some typos and nitpicks:

- In line 1 of page 2, $\|v_{k} \odot\left(K^{\top} u_{k}\right)-q\|$ should be $\|v_{k-1} \odot\left(K^{\top} u_{k}\right)-q\|$. Note that $\|v_{k} \odot\left(K^{\top} u_{k}\right)-q\|$ always equals to 0 due to the Sinkhorn iteration equation.

- In figure 3, it’s better not to use colors to distinguish various methods, as this is not friendly for color-blind people; the text font of labels and axes is too small to read.





**Summary Of The Paper:**

In order to solve large-scale optimal transport problems, the authors propose a Douglas-Rachford (DR) splitting method that solves the original OT without entropy regularization.
The proposed DROT algorithm has the same cost per iteration as the well-known Sinkhorn method but avoids its numerical issues.
Furthermore, the proposed method possesses the strong convergence guarantees for its linear convergence rate; and it can be implemented efficiently using parallel processing on GPUs.
Experiments verify its efficiency and accuracy compared to Sinkhorn.


**Summary Of The Review:**

The proposed method is interesting, while some points listed above need to be clarified.

---

> ### Author Response · Authors · 2021-11-13
> **Response to Reviewer qeU8**
>
> Thank you very much for your positive and constructive feedback! We will make sure to revise according to your comments on typos in upcoming versions.
> 1. **The reviewer suggests performing experiments on much larger datasets.**
>
> In the paper, we have carried out experiments with the dimensions $m=n$ in the range $[100, 200, 500, \ldots, 10000, 20000]$. As pointed out by Reviewer **szP9**, achieving our level of parallelism at $20000 \times 20000$ OT instances is by no means trivial. The next dimensions in this series would be $50000 \times 50000$. However, at this point, the problem data can no longer be fitted in the GPU memory, unless more assumptions are made.
>
> We would also like to take this opportunity to clarify our choice of working with synthetic data. While we can certainly run on some datasets and claim victory, we feel that to support a **hypothesis on robustness**, one really needs a reliable **statistic**. To this end, we solved a total of 300 random problem instances under various settings to produce the performance profile. Such an exhaustive evaluation would be very hard to provide using real datasets, as we were not aware of any comprehensive benchmark suite for OT problems (in the spirit of the CUTEr/st benchmarks for nonlinear programming).
>
> 1. **The reviewer suggested a comparison between our approach and alternative ways of coping with the numerical issues introduced by entropic regularization.**
>
> To our knowledge, all the stable OT solvers up to this date, log-domain stabilization included, come with a huge cost in terms of numerical efficiency, via inefficient memory handling, or poor parallelization. In particular, the documentation of the popular Python package for OT [POT](https://pythonot.github.io/), explicitly warns the user that stabilized Sinkhorn may be extremely slow. To quote the docs:
>
> > If you need to use a small regularization to get sharper OT matrices, you should use the ot.bregman.sinkhorn_stabilized solver that will avoid numerical errors. This last solver can be very slow in practice and might not even converge to a reasonable OT matrix in a finite time.
>
> In addition, we were not aware of any GPU implementation of such stabilized Sinkhorn methods. Therefore, we decided early on to exclude these from our benchmark, as Sinkhorn in its standard form is the main competitor to our kernel implementation in terms of numerical efficiency. We will add a sentence to make this more clear in the final paper.

---

### Official Review · Reviewer_Jdt4 · 2021-11-03

**Correctness:** 3
**Technical Novelty And Significance:** 3
**Empirical Novelty And Significance:** 2
**Recommendation:** 3
**Confidence:** 4

**Main Review:**

The authors address an important problem and come up with an interesting algorithm. They also provide a rigorous theoretical convergence analysis as well as many insights on their CUDA implementation. The performance of the proposed method is competitive with Sinkhorn.

Unfortunately this paper also has some major limitations. The biggest one lies in its rather modest set of experiments. Some fundamentally important experiments are missing to support the claims of the paper.

1. Implementation is presented as a core contribution of this paper: it is mentioned in the title, and in addition, it is devoted a substantial amount of space for presentation (almost 3 pages, a part of Section 3.1 and especially the whole Section 3.3). Therefore, it would be absolutely necessary to show that the presented materials are important by actually comparing the provided implementation (with highly-optimized CUDA kernels) with a naive implementation, e.g., using GPU linear algebra libraries such as PyTorch or TensorFlow. The proposed method (Algorithm 1) can be easily implemented using these libraries in just a few lines.

2. The color transfer experiment is weird. Why aren't the results of Sinkhorn presented? Moreover, if sparsity is considered as an important objective, then the authors should also compare with Blondel et al. (2018). I would suggest to consider a more diverse set of examples to showcase the performance of DR in comparison with Sinkhorn (see also the next point).

3. Sinkhorn has been highly successful not only because of its efficiency, but also because its updates are differentiable, which makes it suitable for learning with stochastic gradient descent. While the updates of Algorithm 1 are also differentiable almost everywhere, it is unclear how this algorithm performs in learning, compared to Sinkhorn. I would encourage the authors to add some learning experiments to make the paper even more solid.


Finally, this paper also has some presentation issues.

1. Major issues:

- I wonder why the authors decided to denote 1_n as "e" and 1_m as "f". The notation "f" is used at multiple locations with different meanings (while the "e" can be confused with the Euler's number, though this is less critical than the "f"). It would be laborious to change because this is used everywhere, so I would suggest (as an acceptable fix) to replace the other instances of "f" with "h", e.g., "h(x) + g(x)" instead of "f(x) + g(x)".

- In Figure 1 and 2, the descriptions of the axes should be clear from the captions (instead of being hidden in the text). Furthermore, the text in the plots (such as axis labels, markers, legends) are unreadable (too small).

- Figure 3b shows that Sinkhorn is actually faster to reach a given accuracy (even though it is less numerically stable). This would deserve some comments (in a fair comparison of two methods, we should present both strengths and weaknesses of both methods).

2. Minor issues:

- Page 2, last paragraph of Section 1.1: "Liang et al. (2017), derived" --> "Liang et al. (2017) derived"

- Page 2, line -5: "methods(Rockafellar, 1976)" --> "methods (Rockafellar, 1976)"

- Section 2 2nd paragraph:
	- "computed as Bauschke et al. (2021):"  --> "computed as (Bauschke et al., 2021):"
	- "linear program on the form" --> "linear program of the form"

- Page 3 last paragraph: "problems on the form" --> "problems of the form"

- Section 3, 2nd paragraph: "and fact that" --> "and the fact that"

- The notation for indicator function in Eq. (7) should be defined.

- Page 5 first paragraph: "matrix-vector multiples" --> "matrix-vector multiplications"

- Page 8 last paragraph: "A algorithm" --> "An algorithm"

---
**Update on November 10th:**

I would like to have some additional comments/questions. Hopefully the authors could address them as well in their rebuttal:

1. There seems to be a minor (likely fixable) **flaw** in Theorem 1 and the discussion thereafter. The inequality
\begin{equation}
\langle C,\bar{X}\_k \rangle - \langle C,X^* \rangle \le \mathcal{O}(\|Y\_0-Y^*\|^2/k)\qquad (\star)
\end{equation}

is **not sufficient** to yield the convergence of the objective (let alone its rate). Note that **$\bar{X}_k$ might not be feasible** and thus $\langle C,\bar{X}\_k \rangle - \langle C,X^* \rangle$ might be negative. In order to obtain the convergence of the objective, one would need a lower bound as well:
\begin{equation}
\alpha_k \le \langle C,\bar{X}\_k \rangle - \langle C,X^* \rangle \le \mathcal{O}(\|Y\_0-Y^*\|^2/k),\qquad (\star\star)
\end{equation}
where $\alpha_k\to 0$. Furthermore, in order to obtain the claimed $\mathcal{O}(1/k)$ rate of convergence, one would also need to show that $(\alpha_k)$ converges to $0$ at the same rate.

2. The authors mentioned that they use the Sinkhorn implementation from POT. It is important to provide enough details so that the interested reader could reproduce the results. For example, which function of POT did the authors use and with which arguments? (ot.sinkhorn has quite several options for its arguments.)

**Summary Of The Paper:**

This paper proposes a novel method for solving the discrete optimal transport problem based on the celebrated Douglas-Rachford splitting (DR) method. The core idea is to split the original transportation map into two variables X and Z, where X is simply non-negative and Z is real-valued and obeys the given row and column sums constraints. The Z-update step in DR is then non-trivial, consisting in projecting a given matrix onto the set of all real-valued matrices with such row/column constraints, which has a closed-form solution due to Bauschke et al. (2021). The authors present a detailed discussion on how to implement the resulting algorithms efficiently on GPUs, and provide a theoretical convergence analysis that recovers the known O(1/k) rate of convergence for DR method. Experiments on a synthetic problem show that the proposed method (using the authors' CUDA implementation) is more numerically stable than the standard Sinkhorn method, while being comparably fast.

**Summary Of The Review:**

The proposed algorithm and analysis/implementation are interesting but the experiments are not convincing. The paper also has some issues with the presentation (but these can be easily fixed).

---

> ### Comment · Reviewer_Jdt4 · 2021-11-10
> **Further comments/questions**
>
> Dear authors,
>
> I have updated my review with some additional comments/questions. I came up with these when reviewing another related paper and was planning to post them during our discussion, but then I thought it would be better to do that as soon as possible so that you can have the time to address them.
>
> Best regards.

---

> > ### Author Response · Authors · 2021-11-12
> > **Response to further comments from Reviewer Jdt4**
> >
> > 1. **The reviewer concerns about the bound on the objective gap.**
> >
> > The reviewer is right that the gap can be negative. However, there is no problem here. First, recall the function $V$ in Theorem 1:
> > $$
> > V(X, Z, U) = f(X) + g(Z) + \langle U, Z-X \rangle.
> > $$
> > Noticing that $V$ is nothing else but the Lagrangian function associated with the composite optimization problem. Primal-dual solutions $(X^\star, U^\star)$ are exactly saddle points of $V$:
> > $$
> > V(X^\star, Z^\star, U)
> >  \leq
> > V(X^\star, Z^\star, U^\star)
> > \leq
> > V(X, Z, U^\star), \quad \forall X \in \mathrm{dom} f, \forall Z \in \mathrm{dom} g.
> > $$
> > Since $X^\star = Z^\star$, letting $X = \bar{X}_k$ and $Z=\bar{Z}_k$, we obtain
> > $$
> > \begin{aligned}
> > V(X^\star, Z^\star, U^\star) &=  \langle C, X^\star \rangle\\\\
> > V(\bar{X}_k, \bar{Z}_k, U^\star) &=\langle C, \bar{X}_k \rangle +
> > \langle U^\star, \bar{Z}_k - \bar{X}_k \rangle.
> > \end{aligned}
> > $$
> > It follows that
> >
> > $$
> > \langle C, \bar{X}_k \rangle - \langle C, X^\star \rangle \geq - \Vert U^\star \Vert \Vert\bar{Z}_k - \bar{X}_k \Vert.
> > $$
> >
> > The second bound in Theorem 1 asserts that $\Vert\bar{Z}_k - \bar{X}_k \Vert \leq O(1/k)$. Therefore, both the upper and lower bounds of the objective gap converge to zero at the same rate $O(1/k)$, as the reviewer desired.
> >
> > While this is a common practice in reporting convergence results of ADMM and splitting-based methods (see., e.g. [Bec17, Theorem 15.4]), we agree that this could cause some confusion and will add a sentence to emphasize further in the final paper.
> >
> > 2. **Sinkhorn parameters detail**
> >
> > Indeed, we specified all the needed information including the number of iterations and stopping criterion. We also updated the section for better readability. Finally, our implementation will be made publicly available for full reproducibility.
> >
> > **Presentation issues**
> >
> > We thank the reviewer for the useful suggestions on notation and presentation. We will carefully use them to improve the paper further.
> >
> > **Final words**
> >
> > We hope that we have addressed all your questions and concerns. Given that you agree with the other reviewers on the novelty and significance of our algorithm design, analysis, and implementation, we hope that you will consider raising your initial score. If you have any outstanding concerns, please let us know, so that we can address those.
> >
> > **References**
> >
> > [Bec17] A. Beck. First-order methods in optimization, volume 25. SIAM, 2017.

---

> > > ### Comment · Reviewer_Jdt4 · 2021-11-15
> > > **Thanks**
> > >
> > > Dear authors,
> > >
> > > Thank you for your feedback.
> > >
> > > Could you please merge this comment with your other comment so that I can give my feedback in a single thread? (You can then safely delete this comment after that.)
> > >
> > > That would make the discussion easier to follow.
> > >
> > > Thank you in advance.
> > > Best regards.

---

> > > > ### Author Response · Authors · 2021-11-15
> > > > **Thanks**
> > > >
> > > > Dear Reviewer Jdt4,
> > > >
> > > > Thank you for reaching out!
> > > >
> > > > We also feel that splitting the comments breaks the flow of the discussion. Unfortunately, there is a word limit for each comment that one can post.
> > > >
> > > > Best regards,
> > > >
> > > > Authors

---

> ### Author Response · Authors · 2021-11-12
> **Response to Reviewer Jdt4**
>
>
> We thank the reviewer for the excellent summary of our contributions and for appreciating the novelty of our work. We address all major concerns below.
>
> 1. **The reviewer concerns whether our kernel is necessary**.
>
> Our best effort using high-level libraries (even CUBLAS) required **11** memory operations on $m \times n$ arrays. As accessing the **global memory** of the GPU requires hundreds of clock cycles, the actual runtime scales almost linearly with the number of such operations. This was indeed the starting point of our tailor-made kernel by which we are able to bring the number down to **2.5**.
> In the paper, we focused on comparing with the Sinkhorn method instead of the suboptimal implementation. We mentioned that the $Y$-update alone would need **8** operations in Section 3.1.
> Nonetheless, we agree with the reviewer that this should be better stressed in the final paper.
>
> Quantitatively, the table below shows the **median** per-iteration times (in miliseconds) of our kernel and a PyTorch implementation for various dimensions $m=n$.
>
> |     |    100      | 200    | 500   | 1000  | 2000 | 5000 | 10000 |
> | :---        |    :----:   |  :---: | :---: | :---: | :---: |:---: |:---: |
> | PyTorch     | 0.809 | 0.827 | 0.84 |  0.984 | 2.18 | 10.852 | 41.192 |
> | Our kernel   | 0.126 | 0.136 | 0.16 | 0.174 | 0.466 | 2.688 | 9.539 |
> | Speedup | 6.439 | 6.064 | 5.266 | 5.663 | 4.676 | 4.037 | 4.318 |
>
> For very large $n$, the relative speedup converges to $11/2.5 \approx 4.4$ as expected.
>
> 2. **Regarding the color transfer example**
>
> We mentioned in the contribution subsection that sparsity is a **by-product** of our approach. The color transfer example is by no means a comparison to other methods but rather an illustration that the splitting method indeed provides sparse solutions. We did not claim any superiority over other algorithms in the related discussion. If the reviewer thinks the illustration is irrelevant, we are happy to move it to the appendix.
>
> 3.  **The reviewer suggests discussing the **learning** aspect of our method.**
>
> To facilitate end-to-end learning, one can solve the regularized objective $\langle C, X \rangle + (\gamma/2) \Vert X \Vert^2$. As in Sinkhorn, a strong convex regularizer ensures the differentiability of the solution. Remarkably, the only change to the original algorithm is just a scalar scaling of the $X$-update:
> $$
> X_{k+1}= [Y_k - \rho C]_{+} \times \frac{1}{1+\rho\gamma}.
> $$
> Our kernel is readily extended to this case, giving an even more flexible algorithm. This should be definitely commented on in the paper.
>
> It is known that the influence of the regularization parameter in Sinkhorn is crucial for the overall learning method [GPC18]. We believe that the accuracy and robustness of our method make it a good candidate for such **end-to-end** learning settings. However, we feel that an extensive and carefully constructed set of experiments would be needed to confirm the hypothesis, which is beyond the scope of this work. We leave the **applications** part of our method as a fascinating direction for future work. We are thankful to the reviewer for the great suggestion!
>
> [GPC18] Aude Genevay, Gabriel Peyré, Marco Cuturi, Learning Generative Models with Sinkhorn Divergences,  *International Conference on Artificial Intelligence and Statistics*, PMLR 84:1608-1617, 2018.

---

> > ### Comment · Reviewer_Jdt4 · 2021-11-15
> > **Thanks for the feedback, but some issues still remain**
> >
> > Dear authors,
> >
> > Thank you for your feedback.
> >
> > > **1. The reviewer concerns whether our kernel is necessary.**
> > Our best effort using high-level libraries...
> >
> > It should be clarified that my concern was not whether your kernel is necessary. It was rather that you didn't show that your kernel was necessary. I hope you see the difference. This is purely an expositional issue. If you spend 3 pages presenting something, you should show that it's worth the space (e.g., showing that it leads to significant gain rather than an incremental improvement of 1% in speed). That's why the comparison with PyTorch above is actually important (in view of the paper's presentation) and should be included in the main content, which would make your paper much more solid.
> >
> > > **2. Regarding the color transfer example**
> > We mentioned in the contribution subsection that sparsity is a by-product of our approach. The color transfer example is by no means a comparison to other methods but rather an illustration that the splitting method indeed provides sparse solutions. We did not claim any superiority over other algorithms in the related discussion. If the reviewer thinks the illustration is irrelevant, we are happy to move it to the appendix.
> >
> > Appendix seems to be a more suitable place for this illustration, but then the main content is left with only synthetic examples, which doesn't look good either. That's why I recommended to do more comparative experiments. See also below.
> >
> > > **3. The reviewer suggests discussing the learning aspect of our method.**
> >
> > Please note that my major concern lies in the modest set of experiments as a whole. In the current manuscript, **the only comparative experiment is one with synthetic data**. In my review I was trying to give the authors ideas to improve their paper in this aspect, and learning experiments were one of the suggestions. These experiments could substantially improve the paper but if they are not doable, then the authors could choose others.
> >
> > At the very least, I would have expected one comparative experiment on real-world data. For example, a comparison of DROT and Sinkhorn on a color transfer dataset should be easy to do because the authors already have the code for this (and then reproducing plots similar to Figures 2-3, but this time for real-world data, should also be easy).
> >
> > > **The reviewer concerns about the bound on the objective gap.**
> > ...Therefore, both the upper and lower bounds... **as the reviewer desired**...
> >
> > I think it would be beneficial for the authors to view reviewers' feedback as a means of improving their paper, rather than mere questions/comments that need to be addressed. That said, don't consider your rebuttal as simply responding to a reviewer's comments or questions. Instead, use your rebuttal to show how you have improved your paper based on those comments. Keep in mind that sometimes reviewers ask questions not because they need answers, but maybe they just want to point out something that is not clear or not correct.
> >
> > For your information, I already knew about the lower bound before posting my comments above, so you don't need to show it to me. You should, however, show it to the future readers of your paper ;)
> >
> > >...While this is a **common practice** in reporting convergence results of ADMM and splitting-based methods (see., e.g. [Bec17, Theorem 15.4]), we agree that this could cause some confusion and will add a sentence to emphasize further in the final paper.
> >
> > **[Updated on 16/11 at 20h30 GMT+1 for clarifications]** Keep in mind that Beck did not claim an $\mathcal{O}(1/k)$ rate of convergence for the *objective* value. The rate in his Theorem 15.4 could be attributed to the convergence of the *residual*. Therefore, the discussion in his book is mathematically correct. By contrast, in the current paper, the authors explicitly claim an $\mathcal{O}(1/k)$ rate of convergence for the objective without providing a lower bound, which is not correct. I have seen multiple papers doing the same, so this is a **common mistake** in mathematical reasoning rather than a common practice.
> >
> > > **Sinkhorn parameters detail**
> > Indeed, we specified all the needed information including the number of iterations and stopping criterion. We also updated the section for better readability. Finally, our implementation will be made publicly available for full reproducibility.
> >
> > Could you share the code snippet where you call `ot.sinkhorn`? As I said, this function has a lot of arguments and the details you provided are not sufficient to know which variant was used.
> >
> > Also, when you make a new revision, **please** highlight the changes (e.g., using a different color). It would be difficult for us reviewers to read the whole paper again to find the changes.
> >
> > --
> >
> > I will increase my score, but at this stage it will unlikely to pass the acceptance bar. Let me further discuss this with my fellow reviewers. (As a reminder: the biggest weakness of this paper lies in the experiments.)

---

> > > ### Comment · Reviewer_Jdt4 · 2021-11-16
> > > **Update for clarifications**
> > >
> > > For information, I have updated a paragraph in my reply above to add clarifications.

---

> > > ### Author Response · Authors · 2021-11-22
> > > **Thank you for your response.**
> > >
> > > Dear Reviewer Jdt4,
> > >
> > > Thank you for your response.
> > >
> > > 1) The reviewer's only remaining concern is the lack of experiments on real-world data.
> > >
> > > We would like to take this opportunity to clarify our choice of working with synthetic data. We felt that to support a **hypothesis on robustness**, one really needs a reliable **statistic**. Moreover, in contrast to, for example deep learning problems, OT is a well-defined class of linear programs with well-defined global optimality properties. We thus focused on synthetic data so that we can easily generate a diverse and large set of OT instances. Such an exhaustive evaluation would be very hard to provide using real datasets, as we were not aware of any comprehensive benchmark suite for OT problems (in the spirit of the CUTEr/st benchmarks for nonlinear programming).
> > >
> > > However, we agree with the reviewer that not every reader is convinced by experiments on synthetic data, and believe that the paper will have a much broader appeal with extra experiments on real-world data. We thus complemented the current results with a new set of experiments on the MNIST data set. We follow the construction in [Cur13] and compare distributions of pixel intensity between images. More concretely, for each pair of source and target digits, we convert an image into a vector of intensity of $28 \times 28$ and normalize it to form the probability vectors $p$ and $q$, respectively. We compared in total 600 random pairs of images to create the performance profile in Figure 3. The new results highlight the effectiveness of our method also in this setting.
> > >
> > > 2) Sinkhorn implementation detail
> > >
> > > We followed the [`sinkhorn_knopp`](https://github.com/PythonOT/POT/blob/master/ot/bregman.py) implementation in POT. This function is the actual implementation of the wrappers `ot.sinkhorn` and `ot.sinkhorn2`. We made one minor modification to `sinkhorn_knopp` to compute the constraint violation, namely $\Vert [X \mathbf{1}_n -p, X^\top \mathbf{1}_m -q]\Vert$. For some reason, the POT toolbox only computes the partial violation $\Vert X^\top \mathbf{1}_m -q\Vert$ as the stopping criterion. We then call the `sinkhorn_knopp` as follows:
> > >
> > > ```Python
> > > sinkhorn_knopp(p, q, C, reg, numItermax=1000, stopThr=1e-4)
> > > ```
> > > Here, $\mathrm{reg} = 1e-4, 1e-3, 5e-3, 1e-2, 5e-2, 1e-1$ is the regularization parameter for `SK1`, `SK2`,..., `SK6`, respectively.
> > >
> > > Finally, thank you for your feedback and sorry for any miscommunication issues that we might have.
> > >
> > > **References**
> > >
> > > [Cur13] M. Cuturi. Sinkhorn distances: Lightspeed computation of optimal transport. Advances in neural information processing systems, 26:2292–2300, 2013.

---

> > > > ### Comment · Reviewer_Jdt4 · 2021-11-29
> > > > **It seems you have used the wrong implementation for the baseline**
> > > >
> > > > Dear authors,
> > > >
> > > > Thanks for the updates.
> > > >
> > > > Unfortunately I'm afraid that you have used the wrong Sinkhorn implementation for the experiments. As far as I know, `sinkhorn_knopp` is the naive variant of Sinkhorn. At the very least, the **standard log-domain variant** (`sinkhorn_log`) should have been used, especially when numerical stability is the main comparative criterion as presented in the paper.
> > > >
> > > > Therefore, the main experiments of this paper are flawed. At this point I'm afraid that you'll have to redo all the experiments. (It is too late now I guess. In my initial review, I explicitly asked for which variant of `ot.sinkhorn` did you use for the experiments, but unfortunately you didn't answer so I couldn't raise this point earlier.)

---

> > > > > ### Author Response · Authors · 2021-11-29
> > > > > **Clarification**
> > > > >
> > > > > First, we stress that we stated explicitly in the paper that we compared with the Sinkhorn method in our experiments. Therefore, we must use the SK implementation instead of log-domain variants.
> > > > >
> > > > > There is currently no method that can solve the exact OT problem to the accuracy and at the speed that we do, neither in theory nor in practice. The experiments that we carry out in the paper demonstrate that the method has the same per-iteration cost as Sinkhorn, and computes accurate results much more reliably than Sinkhorn.
> > > > >
> > > > > There are a number of solvers for approximate OT. Of these, Sinkhorn is by far the fastest one, but it suffers from numerical issues when we decrease the epsilon parameter. Doing computations in log-domain, as you suggest, is one of many approaches to try to increase the range of epsilons that can be handled, but it results in a much slower solver (at least a factor 3x or 4-5x if stopping criterion is evaluated). This is because the log domain implementation requires $m\times n$ computations of the exponential function and **line-by-line** soft-minima of the resulting matrices in each step, losing all the benefits of fast matrix-vector multiplications in SK [see, e.g. PC19]. If we were to compare with the log-domain implementation, we would not have needed to design our kernel in the first place. Another example is the **stabilized_sinkhorn** method, which implements a more carefully optimized version of the above method, which we discussed in detail in the response to Reviewer qeU8.
> > > > >
> > > > > As we strike for the minimal cost per iteration of Sinkhorn in our work, we focused on its fastest and most widely used version. We agree that adding comparisons with more SK versions, such as the one you suggested, would make the paper even more solid. We however disagree that our experiments are flawed.
> > > > >
> > > > > [PC19] G. Peyré and M. Cuturi. Computational optimal transport. *Foundations and Trends in Machine Learning*, 11(5-6):355–607, 2019.

---

### Author Response · Authors · 2021-11-12
**General comments to all the reviewers**

We thank the reviewers for their careful reading and the
valuable feedback of our work. We were happy to hear the overall positive nature of the reviews and the constructive
suggestions provided.

We have made the following changes to the submitted version of the paper:
- In Theorems 1 and 2, we have replaced the big-O notation with explicit constants as they are indeed very neat.
- The experimental section has been expanded with a new set of experiments on a real data set.
- The abstract and contributions have been updated to better stress the contributions in convergence rates.

Before addressing each reviewer’s comments, we would like to reinforce that our work contributes significantly on **three fronts**:
- **Algorithm design**: We develop a completely new and efficient splitting algorithm for OT
- **Convergence analysis**:  We establish a significantly better convergence rate than the Sinkhorn method and a linear rate, despite the lack of strong convexity.
- **Implementation**: We develop a high-performance CUDA kernel for the proposed method.

These make operator splitting methods, for the first time, truly competitive for solving large-scale OT, both in theory and practice.

---

### Decision · Program_Chairs · 2022-01-20

**Decision:**

Accept (Poster)

**Comment:**

This paper proposes a new algorithm to solve the discrete optimal transport problem, consisting of showing how the well-known Douglas Rachford algorithm can be efficiently applied, and also providing a convergence rate. Secondly, the paper gives an efficient implementation suitable for GPUs.

The paper would be a bit weak on just the DR contribution alone, so the efficient implementation, and experiments, are significant. However, the reviewers were not consistently happy with the experiments.

Two big issues raised by reviewers were wanting more experiments (having comparison experiments with real-world data), and wanting to compare with variants of the baseline algorithm (Sinkhorn), such as the log-transformed version which is more stable.  Looking at the revision, I think the reviewers have done a good job adding experiments.  Overall, for a paper with strong theoretical components, I think the computational aspects are strong.

As for (not) comparing with the log-transformed Sinkhorn and other more robust methods, the authors argue that this implementation would be slower.  I agree with reviewers that it would be nice to have these comparisons, but the authors' argument is plausible and I don't find it grounds to reject the paper.

Overall, it seems there is some evidence that this is a worthwhile method, and there are no theoretical concerns other than presentation issues. Thus I think it would be a benefit to the community to accept this paper.